# Male meiotic spindle features that efficiently segregate paired and lagging chromosomes

Gunar Fabig[1]*, Robert Kiewisz[1], Norbert Lindow[2], James A Powers[3], Vanessa Cota[4], Luis J Quintanilla[4], Jan Brugués[5,6,7], Steffen Prohaska[2], Diana S Chu[4†], Thomas Müller-Reichert[1†]*

[1]Experimental Center, Faculty of Medicine Carl Gustav Carus, Technische Universität Dresden, Dresden, Germany; [2]Zuse Institute Berlin, Berlin, Germany; [3]Light Microscopy Imaging Center, Indiana University, Bloomington, United States; [4]Department of Biology, San Francisco State University, San Francisco, United States; [5]Max Planck Institute of Molecular Cell Biology and Genetics, Dresden, Germany; [6]Max Planck Institute for the Physics of Complex Systems, Dresden, Germany; [7]Centre for Systems Biology Dresden, Dresden, Germany

*For correspondence:
gunar.fabig@tu-dresden.de (GF);
mueller-reichert@tu-dresden.de
(TMü-R)

†These authors contributed equally to this work

Competing interests: The authors declare that no competing interests exist.

**Abstract** Chromosome segregation during male meiosis is tailored to rapidly generate multitudes of sperm. Little is known about mechanisms that efficiently partition chromosomes to produce sperm. Using live imaging and tomographic reconstructions of spermatocyte meiotic spindles in *Caenorhabditis elegans*, we find the lagging X chromosome, a distinctive feature of anaphase I in *C. elegans* males, is due to lack of chromosome pairing. The unpaired chromosome remains tethered to centrosomes by lengthening kinetochore microtubules, which are under tension, suggesting that a 'tug of war' reliably resolves lagging. We find spermatocytes exhibit simultaneous pole-to-chromosome shortening (anaphase A) and pole-to-pole elongation (anaphase B). Electron tomography unexpectedly revealed spermatocyte anaphase A does not stem solely from kinetochore microtubule shortening. Instead, movement of autosomes is largely driven by distance change between chromosomes, microtubules, and centrosomes upon tension release during anaphase. Overall, we define novel features that segregate both lagging and paired chromosomes for optimal sperm production.

## Introduction

Chromosome segregation during meiosis is regulated in each sex to produce different numbers of cells with distinct size, shape, and function. In humans, for example, up to 1500 sperm are continually generated per second via two rapid rounds of symmetric meiotic divisions. In contrast, in many organisms including humans, only oocytes that are fertilized will complete asymmetric meiotic divisions to produce one large cell and two polar bodies (*El Yakoubi and Wassmann, 2017*; *L'Hernault, 2006*; *O'Donnell and O'Bryan, 2014*; *Severson et al., 2016*; *Shakes et al., 2009*). While oocyte meiosis and mitosis have been studied in detail in many organisms (*Bennabi et al., 2016*; *Müller-Reichert et al., 2010*; *Pintard and Bowerman, 2019*), our knowledge of sperm meiotic chromosome segregation is still limited to studies in grasshoppers and crane flies using chromosome manipulation and laser microsurgery (*LaFountain et al., 2011*; *LaFountain et al., 2012*; *Nicklas and Kubai, 1985*; *Nicklas et al., 2001*; *Zhang and Nicklas, 1995*). Thus, despite recent alarming evidence of steep global declines in human sperm counts (*Levine et al., 2017*; *Levine et al., 2018*; *Sengupta et al., 2018*), little is known about the molecular mechanisms that drive male meiotic chromosome segregation required for efficiently forming healthy sperm.

The nematode *Caenorhabditis elegans* is an ideal model system to study sperm-specific features of chromosome segregation, as meiosis can be visualized in both sexes. *C. elegans* lacks a Y chromosome; thus, sex is determined by X chromosome number. Hermaphrodites have two X chromosomes (XX), while males have one (XO). This unpaired (univalent) X chromosome lags during anaphase I in males (*Albertson and Thomson, 1993*; *Fabig et al., 2016*; *Madl and Herman, 1979*). Previously, electron microscopy has defined the microtubule organization in female meiotic (*Laband et al., 2017*; *Redemann et al., 2018*; *Srayko et al., 2006*; *Yu et al., 2019*) and embryonic mitotic spindles (*Albertson, 1984*; *O'Toole et al., 2003*; *Redemann et al., 2017*; *Yu et al., 2019*) but a detailed study on spindle ultrastructure in spermatocytes, particularly at anaphase I showing the lagging X chromosome, is lacking.

Compared to oocytes or mitotic embryonic cells in *C. elegans*, spermatocytes exhibit vastly distinct features of spindle poles, chromosomes, and kinetochores (*Crowder et al., 2015*; *Hauf and Watanabe, 2004*). As in many other species, *C. elegans* centrosomes are present in spermatocyte meiosis (*Wolf et al., 1978*) and embryonic mitosis (*O'Toole et al., 2003*) but not in oocyte meiosis, where inter-chromosomal microtubules are reported to push chromosomes apart (*Dumont et al., 2010*; *Laband et al., 2017*; *Redemann et al., 2018*; *Yu et al., 2019*). Moreover, meiotic chromosomes in *C. elegans* resemble compact oblong spheres in spermatocytes and oocytes (*Albertson and Thomson, 1993*; *Redemann et al., 2018*; *Shakes et al., 2009*) but are long rods in mitosis (*Oegema et al., 2001*; *Redemann et al., 2017*). In addition, previous studies in *C. elegans* revealed the holocentric nature of meiotic and mitotic kinetochores (*Albertson and Thomson, 1993*; *Howe et al., 2001*; *O'Toole et al., 2003*) and the rounded structure of meiotic chromosomes (*Dumont et al., 2010*; *Monen et al., 2005*; *Muscat et al., 2015*; *Wignall and Villeneuve, 2009*). Meiotic kinetochore structure and dynamics, however, vary from mitosis, where kinetochores attach each sister to microtubules from opposite poles during the single division. Meiotic kinetochores must detach and re-attach to microtubules to allow sisters to switch from segregating to the same pole in meiosis I to opposite poles in meiosis II (*Petronczki et al., 2003*). In acentrosomal oocyte meiosis, this is accomplished because outer kinetochore levels dramatically decrease during anaphase I and then increase again before meiosis II (*Dumont and Desai, 2012*; *Dumont et al., 2010*). The structure and dynamics of centrosomal spermatocyte meiotic spindles, however, are largely unknown.

Different cell types also use distinct spindle structures to drive chromosome movement (*McIntosh, 2017*; *McIntosh et al., 2012*). For example, during the first centrosomal mitotic division in *C. elegans*, pole-to-pole separation (anaphase B) but not chromosome-to-pole shortening (anaphase A) drives chromosome movement (*Nahaboo et al., 2015*; *Oegema et al., 2001*; *Scholey et al., 2016*). In *C. elegans* acentrosomal oocyte spindles, shortening of the distance between chromosomes and poles was observed before microtubules disassemble at the acentrosomal poles (*McNally et al., 2016*). Pushing forces generated by microtubules assembled in the spindle midzone then drive the majority of segregation in oocyte meiosis (*Laband et al., 2017*; *Yu et al., 2019*). As yet, mechanisms that drive segregation in sperm centrosomal meiotic spindles are unknown.

To better understand sex-specific regulation of meiotic chromosome segregation and the resolution of lagging chromosomes, we quantitatively characterized the three-dimensional (3D) organization of spindles and the dynamics of chromosomes in *C. elegans* male spermatocytes. We applied electron tomography to produce large-scale 3D reconstructions of whole spindles in combination with a newly developed light microscopic approach for imaging chromosome and spindle dynamics in living males. Our approach defines molecular mechanisms of sperm-specific movements, focusing on the efficient segregation of both lagging and paired chromosomes.

## Results

### Spermatocyte meiotic spindles are distinguished by delayed segregation of the unpaired X chromosome

We developed in situ imaging within *C. elegans* males to visualize the dynamics of microtubules and chromosomes labeled with β-tubulin::GFP and histone::mCherry, respectively (*Figure 1*). Spermatocyte chromosomes arrange in a rosette pattern, with paired autosomes surrounding the unpaired X chromosome in metaphase I (*Albertson and Thomson, 1993*). In anaphase I, homologs segregate

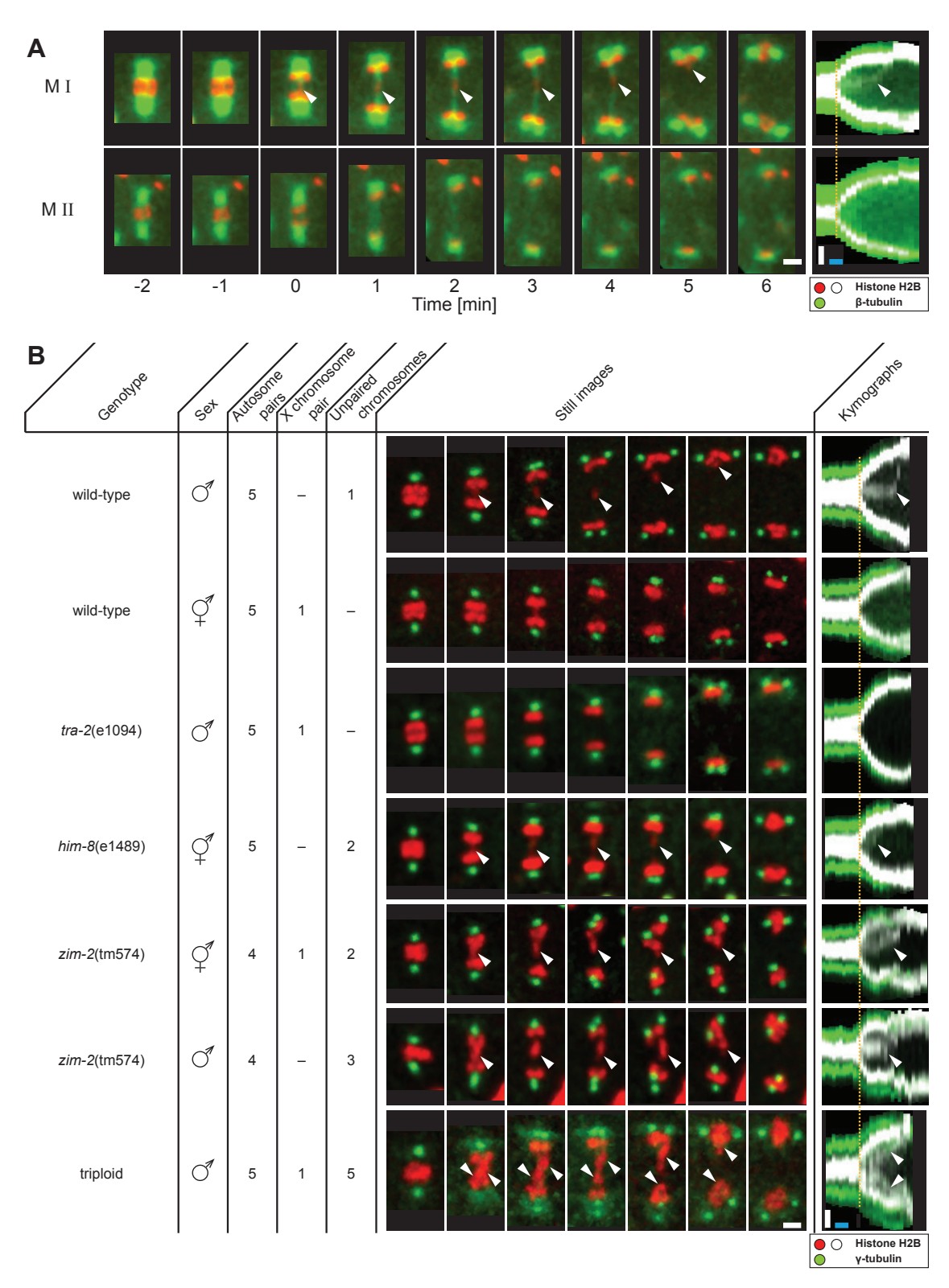

**Figure 1.** Unpaired chromosomes lag in spermatocyte meiosis I. (**A**) Time series of confocal image projections of meiosis I (M I) and meiosis II (M II) in males with microtubules (β-tubulin::GFP, green) and chromosomes (histone H2B::mCherry, red). Anaphase onset is time point zero (t = 0). White arrowheads mark the unpaired X chromosome position. Right panels show corresponding kymographs with chromosomes in white, microtubules in green. Anaphase onset is marked with a dashed orange line. Scale bars (white), 2 μm; time bar (blue), 2 min. (**B**) Confocal image projections of

*Figure 1 continued on next page*

Figure 1 continued

spermatocyte meiosis I in wild-type XO males, wild-type XX hermaphrodites, *tra-2*(e1094) XX males, *him-8*(e1489) XX hermaphrodites, *zim-2*(tm574) XX hermaphrodites, *zim-2*(tm574) XO males and triploid XXO males with centrosomes in green (γ-tubulin::GFP) and chromosomes in red (histone H2B::mCherry). The genotype, sex, number of autosome or X chromosome pairs, and number of unpaired chromosomes is indicated. Still images illustrate the progression of the first meiotic division over time with lagging chromosomes indicated by white arrowheads. In the corresponding kymographs (right panels), chromosomes are shown in white, spindle poles in green. Anaphase onset is marked with a dashed line (orange). Scale bars (white), 2 µm; time bar (blue), 2 min.

The online version of this article includes the following video for figure 1:

**Figure 1—video 1.** Live-cell imaging of meiosis I and II in wild-type males.

https://elifesciences.org/articles/50988#fig1video1

towards opposite poles; the unpaired X chromosome, however, remains behind and attached to microtubules connected to separating poles before resolving to one side (*Figure 1A*, M I, arrowheads, *Figure 1—video 1*; *Albertson and Thomson, 1993*). Thus in meiosis I, the unpaired X appears attached to both poles in contrast to the paired autosomes, each of which attaches to a single, opposite pole to enable segregation of homologs. In the second division, sister chromatids of each chromosome segregate all away from one another to opposite poles (*Figure 1A,M II*).

## Lagging of chromosomes is a consequence of a lack of pairing

Next, we probed if the lagging of X may be due to a lack of having a pairing partner. Because both males and hermaphrodites undergo spermatogenesis in *C. elegans*, we compared spermatocytes of wild-type males (XO) to those in animals with different numbers of chromosomes (*Figure 1B*). First, though the unpaired X chromosome lags in wild-type XO males, paired X chromosomes in wild-type XX hermaphrodite spermatocytes did not. Although we noticed an initial delay in the segregation of a chromosome in hermaphrodite spermatocytes (n = 5/10), this slight lagging was not obvious at mid to late anaphase I. Further, we determined whether paired X chromosomes lag in males by analyzing mutants with the *tra-2*(e1094) mutation, which causes a somatic transformation of XX animals to males (*Hodgkin and Brenner, 1977*). In over 80% (n = 43/53) of *tra-2*(e1094) XX male spindles we did not detect lagging chromosomes during meiosis I. However, in about 20% of *tra-2*(e1094) spindles, we detected lagging to some extent, possibly due to improper pairing in prophase. Therefore, the majority of paired X chromosomes in male spermatocyte spindles do not lag in mid/late anaphase I, similar to paired sex chromosomes in hermaphrodite spermatocytes.

We next examined X chromosome lagging in *him-8*(e1489) hermaphrodite spermatocytes to eliminate the possible effect of the male soma in causing chromosomes to lag in meiosis I. A mutation of *him-8* results in lack of pairing of the X; thus, pairing, synapsis, and recombination of the X chromosomes do not occur (*Phillips et al., 2005*). We observed lagging chromosomes in anaphase I in 70% (n = 14/20) of the analyzed *him-8*(e1489) spindles in hermaphrodite spermatocytes, presumably representing the two unpaired X chromosomes. This reveals that anaphase I chromosome lagging is likely caused by an inability to undergo synapsis rather than by a somatic effect of the male sex.

We further excluded that lagging is exclusive to the X chromosome by analyzing hermaphrodite and male spermatocytes with the *zim-2*(tm574) mutation, which prevents pairing of autosome V (*Phillips and Dernburg, 2006*). At least one chromosome lagged in all spindles in hermaphrodite (n = 10) and male spermatocytes (n = 5). Moreover, we created triploid males with

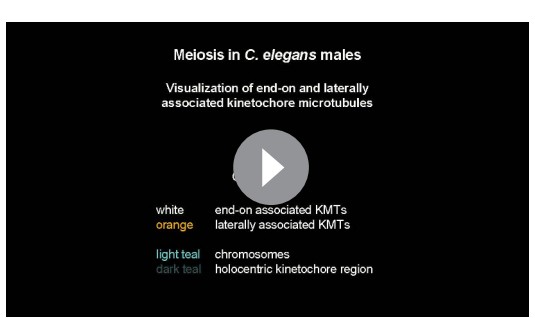

**Video 1.** Visualization of end-on or laterally associated kinetochore microtubules. This video illustrates the classification of kinetochore microtubules according to their type of association to chromosomes. Chromosomes are shown in light teal, the holocentric kinetochore area surrounding the chromosomes in dark semi-transparent teal. End-on associated kinetochore microtubules are shown in white, laterally associated microtubules in orange).

https://elifesciences.org/articles/50988#video1

spermatocytes containing five unpaired autosomes (*Madl and Herman, 1979*) and detected a massive fluorescent signal between segregating autosomes that we infer likely corresponds to the five unpaired autosomes in all spindles (n = 32). Collectively, these results show that lagging chromosomes during spermatocyte anaphase I are indeed a consequence of the lack of pairing and synapsis of any chromosome during prophase I and are not specific to sex chromosomes.

## Microtubules attached to the X chromosome exert a pulling force

During anaphase I, we observed that the lagging X changes shape (*Figure 2A*, upper panel). To quantify this change, we calculated a shape coefficient (the ratio of length over diameter) of the X chromosome in fluorescent 3D image data over time (*Figure 2A*, lower panel). With this measure, stretched chromosomes have a shape coefficient greater than 1. Indeed, the X chromosome was significantly stretched early in anaphase I with a shape coefficient of 1.4 that decreased to 1.0 as it rounded up in late anaphase I (*Figure 2B*). This suggests that the X is under tension from pulling forces as spindle poles separate, which is released as the lagging chromosome resolves to one side.

To further assess pulling forces during anaphase, we used laser microsurgery on X chromosome-attached microtubules. We reasoned a cut on one side of the lagging X would release tension and induce segregation to the opposite side. Laser point-ablation to the bundle on one side of the X caused an immediate and continuous movement of the X towards the unablated side (n = 21/26, two examples shown; *Figure 2C*, *Figure 2—video 1*). The velocity of the X chromosome movement after the cut was variable, similar to unperturbed spindles, which also displayed variability in the speed of X chromosome movement, ranging from 0.7 to 4.9 µm/min (mean = 2.2 µm/min, n = 40) and the onset of X chromosome movement relative to anaphase onset, ranging from 1.5 to 8.5 min (mean = 4.9 min, n = 55, data not shown). We also tested cutting the microtubule bundles sequentially on each side of the X chromosome (*Figure 2D*, *Figure 2—video 2*). After initiation of movement by the first cut, the second cut on the opposite side caused a rapid shift in the direction of segregation, indicating microtubule connections are highly dynamic during anaphase I. Taken together, we conclude that kinetochore microtubules exert a pulling force on chromosomes during anaphase. This tension is most obvious on the microtubules connected to the X chromosome.

## Kinetochores are not disassembled between spermatocyte meiotic divisions

A critical connection between chromosomes and microtubules required for forces that pull chromosomes during anaphase are kinetochores. However, in oocyte meiosis the localization of outer kinetochore proteins on chromosomes drops dramatically during anaphase I, allowing central spindle components to largely drive chromosome movement (*Dumont and Desai, 2012*; *Dumont et al., 2010*). We thus applied immunofluorescence microscopy to determine the localization of kinetochore components. Strikingly, the outer kinetochore proteins, KNL-1 (*Desai et al., 2003*), KNL-3 (*Cheeseman et al., 2004*) and NDC-80 (*Desai et al., 2003*) are retained on the poleward sides of the rounded chromosomes in a cup-shaped pattern on autosomes and X chromosome during anaphase I (*Figure 2E* and *Appendix 1—figure 1*), in contrast to their rapid depletion during oocyte meiosis. Using super-resolution microscopy, we further found kinetochores bridge interactions between chromosomes and microtubules (*Appendix 1—figure 1*). Outer kinetochore retention suggests that chromosome-to-pole attachments are important to drive chromosome movement during sperm anaphase.

## Spermatocyte spindles maintain both end-on and lateral associations of kinetochore microtubules to chromosomes throughout meiosis

To further determine how spermatocyte spindles reorganize during anaphase, we applied large-scale electron tomography to visualize the ultrastructure of whole spindles in different stages of meiosis I with single-microtubule resolution (*Redemann et al., 2017*; *Redemann et al., 2018*; *Yu et al., 2019*). We segmented centrioles, microtubules, autosomes (a), and the X chromosome (x) (*Figures 3* and *4*, left panels). Reconstructed spindles were staged by 1) correlating the pole-to-pole and the pole-to-autosome distance with the autosome-to-autosome distance, 2) comparing our tomographic data with live-imaging data of chromosome and pole dynamics (*Appendix 1—figure 2*), and 3) correlating centrosome dynamics with those observed by live imaging, where centrosome volume

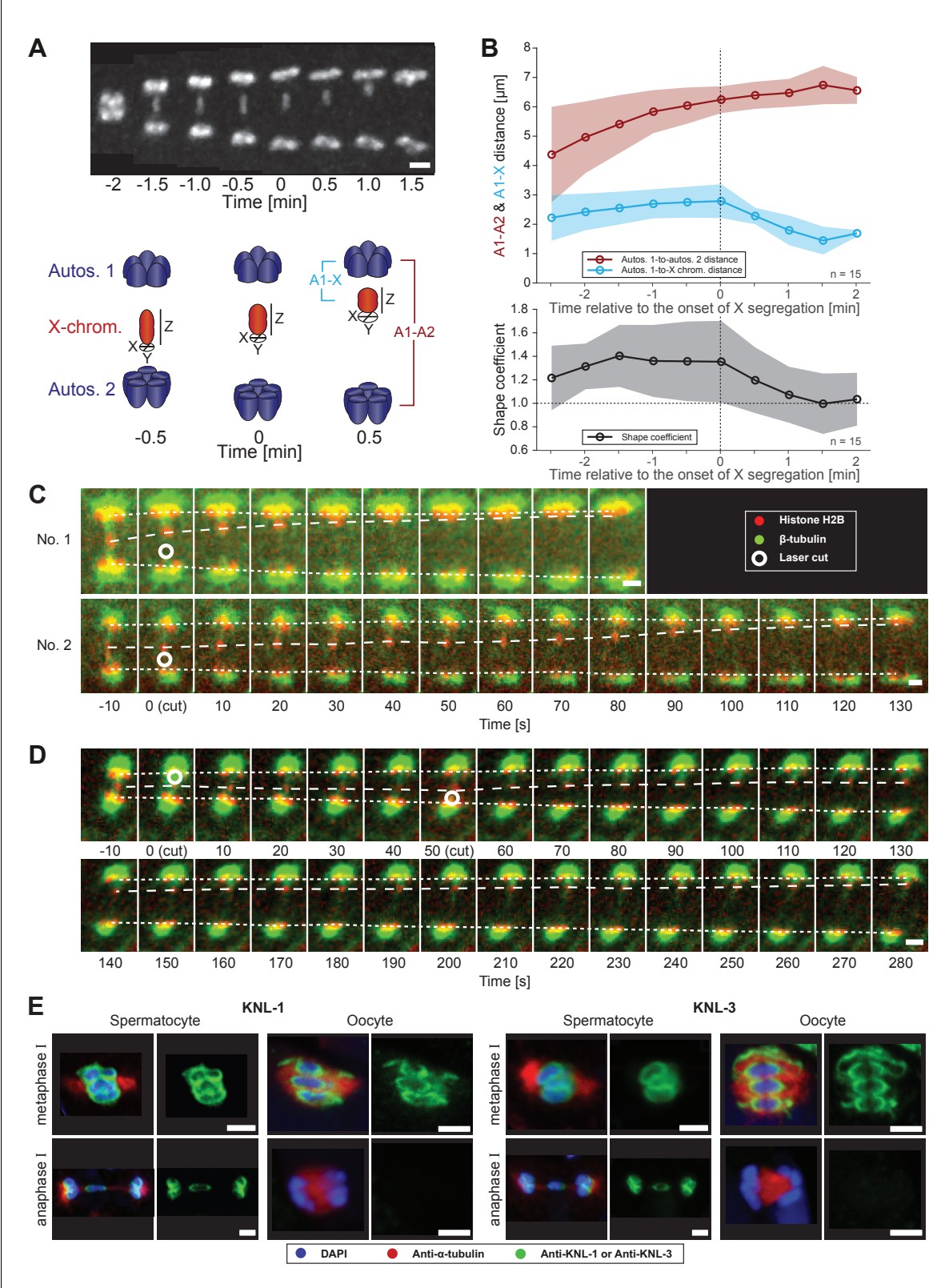

**Figure 2.** Microtubules associated with the X chromosome exert a pulling force. (**A**) Maximum intensity projection images of chromosomes labeled with histone H2B::mCherry (upper panel, white). Time is relative to the onset of segregation of the X chromosome (t = 0). Scale bar, 2 µm. Schematic diagram illustrating the quantification of X chromosome shape (lower panel). The length (Z) of the X chromosome (red) is divided by its width (X + Y divided by two). Autosomes are in blue. (**B**) Plots showing segregation distances and the shape of the X chromosome in anaphase I spindles (n = 15).
*Figure 2 continued on next page*

*Figure 2 continued*

The upper panel shows the autosome 1-to-autosome 2 (A1-A2, red) and the autosome 1-to-X chromosome distances (A1-X, blue) over time, the lower panel shows the shape coefficient (black). Solid lines show the mean, shaded areas indicate the standard deviation. The onset of X chromosome movement is given as time point zero (t = 0). (C) Laser microsurgery of microtubules associated with the X chromosome in anaphase I. Microtubules are labeled with β-tubulin::GFP (green) and chromosomes with histone H2B::mCherry (red). The position of the cut is indicated (white circle). Time is relative to the time point of the applied laser cut (t = 0). The position of the autosomes (outer dashed lines) and the X chromosome (inner dashed line) is indicated. The two panels show examples of X chromosome segregation to the applied cut. Scale bars, 2 μm. (D) Example of a double cut experiment over ~300 s. The two cuts are indicated (white circles). Scale bar, 2 μm. (E) Localization of kinetochore proteins in spermatocyte and oocyte meiosis I. Metaphase (upper row in each panel) and anaphase (lower row in each panel) of the first division is shown. Left panels show whole spindles from fixed males stained with antibodies against the kinetochore proteins KNL-1 or KNL-3 (green), microtubules (red), and DAPI (blue). Right panels show the localization patterns of the kinetochore protein only. Scale bars, 2 μm.

The online version of this article includes the following video and source data for figure 2:

**Source data 1.** Segregation distances and the shape of the X chromosome in replicates of anaphase I spindles analyzed in *Figure 2B* .
**Figure 2—video 1.** Laser ablation of the microtubule bridge in a spermatocyte undergoing the first meiotic division.
https://elifesciences.org/articles/50988#fig2video1
**Figure 2—video 2.** Double-cut laser ablation of the microtubule bridge in a spermatocyte undergoing first meiotic division.
https://elifesciences.org/articles/50988#fig2video2

decreased rapidly after onset of anaphase I and centrosomes flattened out before splitting into two spindle poles (*Schvarzstein et al., 2013*; *Appendix 1—figure 3*; see also Materials and methods). Accordingly, we generated: three spindles at metaphase (*Figure 3A–C*; *Figure 3—videos 1–3*), one at anaphase onset (*Figure 4A*; *Figure 4—video 1*), three complete spindles (*Figure 4B–D*; *Figure 4—videos 2–4*) and three partial spindles (*Appendix 1—figure 4*) at early to mid-anaphase, and one spindle at late anaphase (*Figure 4E*; *Figure 4—video 5*).

In our tomographic reconstructions, the holocentric kinetochore was visible as a ribosome-free zone around the chromosomes (*Howe et al., 2001*; *O'Toole et al., 2003*; *Redemann et al., 2017*) from metaphase throughout anaphase. This is in accordance with kinetochore retention during anaphase I and the association distance measured by super-resolution light microscopy (*Figure 2E* and *Appendix 1—figure 1*). Therefore, microtubules terminating or traversing in these ribosome-free zones with a width of 150 nm were considered kinetochore microtubules. The remaining microtubules were annotated as other microtubules (*Figures 3* and *4* and *Appendix 1—figure 4*, mid left panels; mid right panels showing kinetochore microtubules only).

Next, we determined the number of kinetochore microtubules. We reconstructed ~2000 microtubules in metaphase I and ~1500 in anaphase I data sets. For metaphase I,~30–38% were kinetochore microtubules, which increased to ~45–53% for anaphase I (*Table 1*). Interestingly, this percentage is much higher compared to early mitosis where only 2–4% of all microtubules attach to the kinetochore (*Redemann et al., 2017*). We also determined the types of association kinetochore microtubules make to chromosomes (see Materials and methods). Similar to previous analysis of oocyte meiosis (*Dumont et al., 2010*; *Laband et al., 2017*; *Muscat et al., 2015*; *Redemann et al., 2018*), we recognized both a lateral and an end-on association of kinetochore microtubules to chromosomes (for details see *Table 1*). We found both types of association for the autosomes and the unpaired X at all reconstructed meiotic stages. This indicated to us that a complex pattern of both lateral and end-on associations is maintained throughout male meiotic progression.

## Continuous and lengthening microtubules connect the X chromosome to centrosomes during anaphase I

One phenomenon of spermatocyte meiosis is the kinetochore microtubules that connect the lagging X to opposite spindle poles lengthen during anaphase I. This is unusual because in most centrosomal cell-types, microtubules either shorten (anaphase A) or stay the same length as poles separate (anaphase B). Further, in *C. elegans* mitosis, continuous microtubules do not directly connect centrosomes and chromosomes, but instead anchor into the spindle network (*Redemann et al., 2017*). We thus used electron tomography to determine the continuity of X-connected kinetochore microtubules during anaphase I. We found that microtubules directly connect the X to each centrosome throughout anaphase I (*Figures 5A–B* and *4B–E*, mid right panels, *Appendix 1—figure 5A and B*). Further, microtubules with both end-on and lateral associations to X increased in length as X

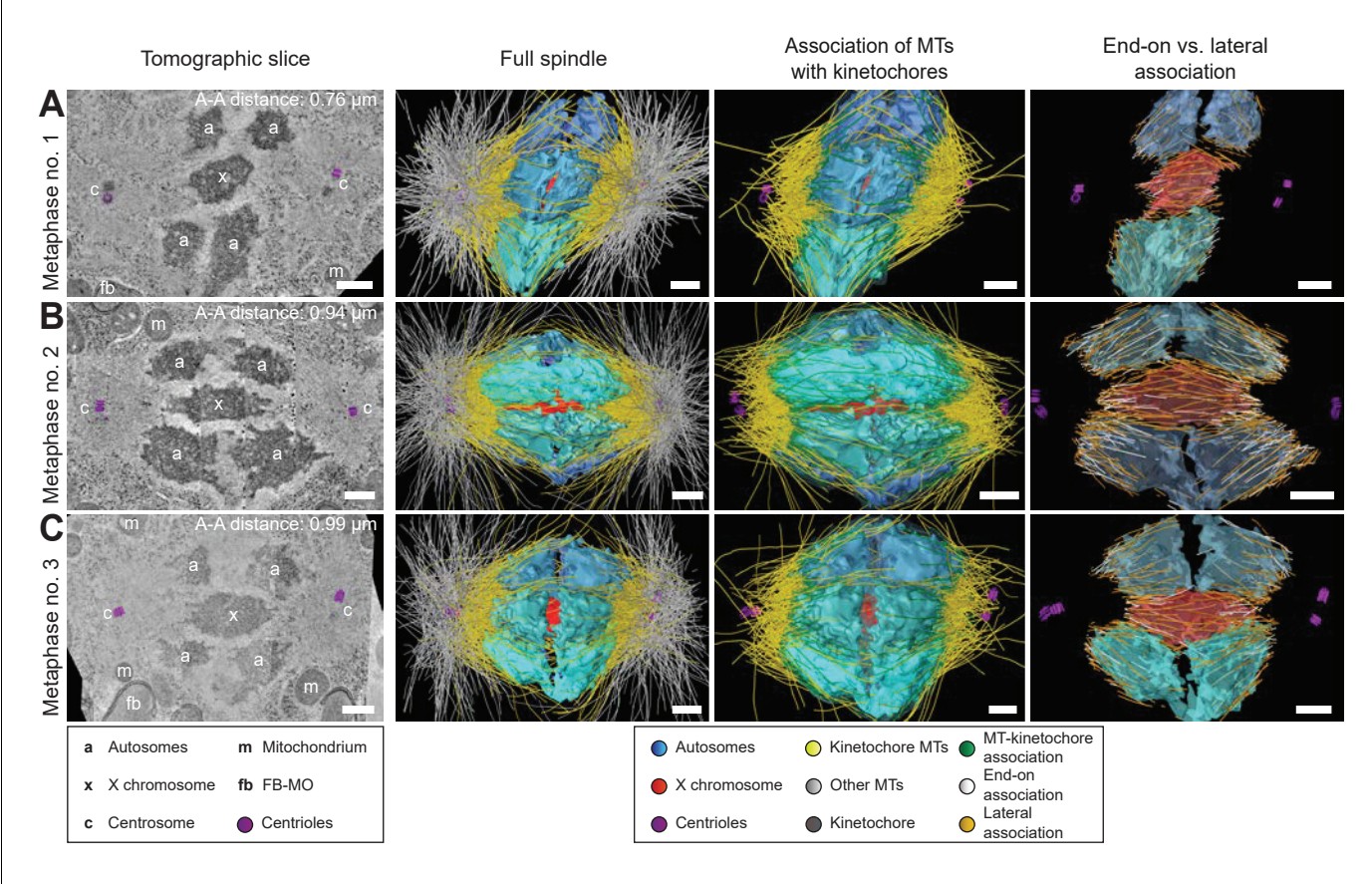

**Figure 3.** Three-dimensional ultrastructure of spindles in metaphase I. (**A**) Early metaphase spindle (Metaphase no. 1) with unstretched chromosomes. (**B–C**) Metaphase spindles (Metaphase no. 2 and 3) with stretched chromosomes. Left panels: tomographic slices showing the centrosomes (c, with centrioles in purple), the autosomes (a), and the unpaired X chromosome (x) aligned along the spindle axis, mitochondria (m) and fibrous body-membranous organelles (fb). Mid left panels: corresponding three-dimensional models of the full spindles. Autosomes are in different shades of either blue or cyan, the X chromosome in red, centriolar microtubules in purple, microtubules within 150 nm to the chromosome surfaces in yellow, and all other microtubules in gray. Mid right panels: association of kinetochore microtubules with the kinetochores. Kinetochores are shown as semi-transparent regions around each chromosome. The part of each microtubule entering the kinetochore region around the holocentric chromosomes is shown in green. Right panels: visualization of end-on (white) *versus* lateral (orange) associations of microtubules with chromosomes. Only the parts of microtubules inside of the kinetochore region are shown. The autosome-to-autosome distance (A-A) for each reconstruction is indicated in the left column. Scale bars, 500 nm.

The online version of this article includes the following video(s) for figure 3:

**Figure 3—video 1.** Full tomographic reconstruction of the metaphase I spindle in a wild-type male spermatocyte.
https://elifesciences.org/articles/50988#fig3video1

**Figure 3—video 2.** Full tomographic reconstruction of the metaphase I spindle in a wild-type male spermatocyte.
https://elifesciences.org/articles/50988#fig3video2

**Figure 3—video 3.** Full tomographic reconstruction of the metaphase I spindle in a wild-type male spermatocyte.
https://elifesciences.org/articles/50988#fig3video3

chromosome-to-pole distance increased until the X resolved to one side (*Figure 5C–D*). Thus, continuous microtubules connect the X chromosome to poles even as poles elongate during anaphase I, consistent with the observation that outer kinetochore protein levels are retained on the X throughout spermatocyte divisions (*Figure 2E*, *Appendix 1—figure 1*).

We also observed that X-associated microtubules were curved during late anaphase I. To assess the curvature of end-on and lateral X-associated kinetochore microtubules, we measured the tortuosity of individual microtubules by calculating the ratio of the spline length over the end-to-end length (*Appendix 1—figure 6A*). At metaphase I and anaphase I onset, the tortuosity ratio was one, indicating kinetochore microtubules were straight. However, at anaphase, X-connected microtubules

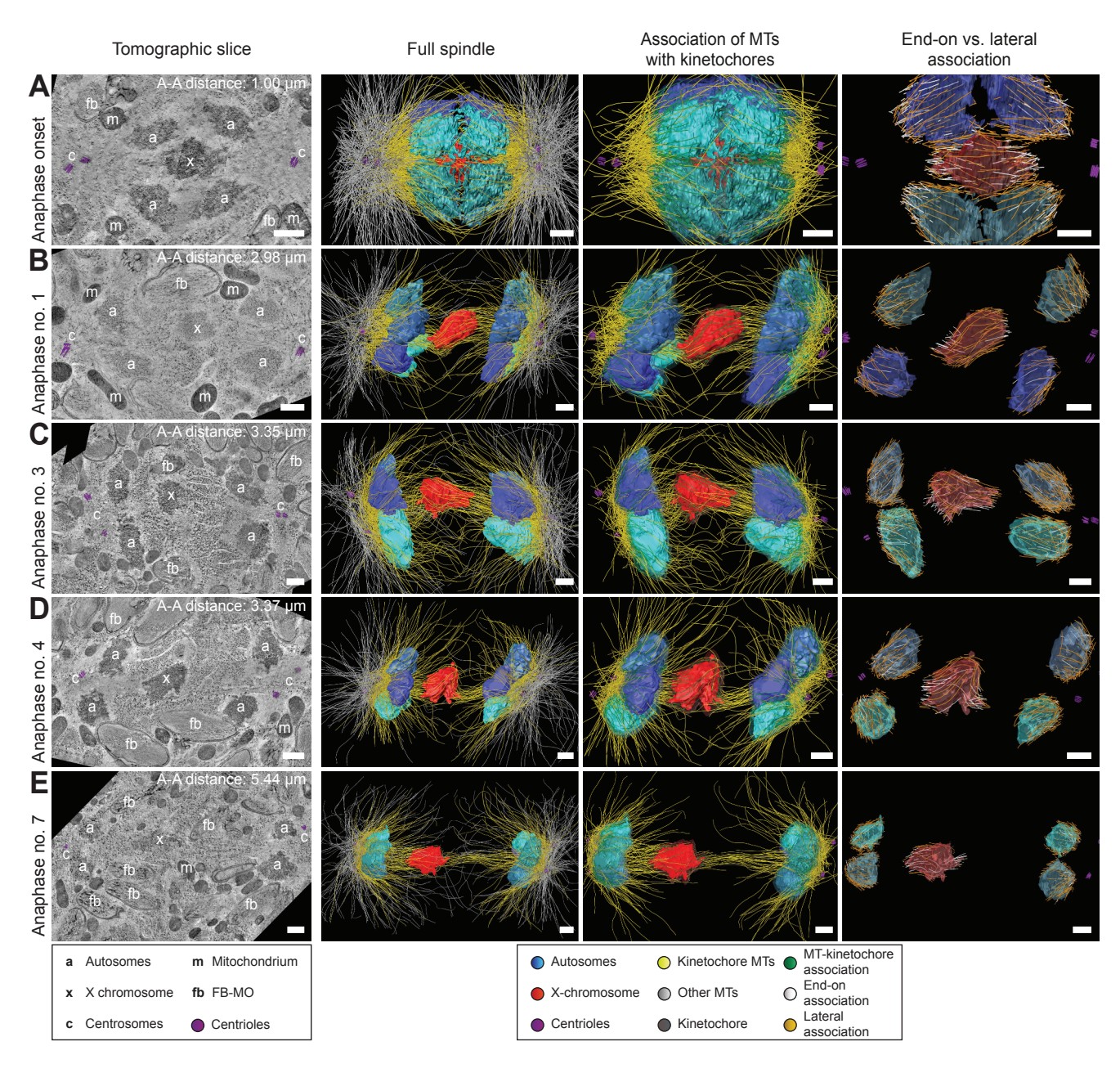

**Figure 4.** Three-dimensional ultrastructure of spindles in anaphase I. (**A**) Spindle at Anaphase onset. (**B**) Spindle at mid anaphase (Anaphase no. 1) with a pole-to-pole distance of 2.98 μm. (**C**) Mid anaphase spindle (Anaphase no. 3) with a pole-to-pole distance of 3.35 μm. (**D**) Mid anaphase spindle (Anaphase no. 4) with a pole-to-pole distance of 3.37 μm. (**E**) Spindle at late anaphase (Anaphase no. 7) with a pole-to-pole distance of 5.44 μm and the X chromosome with initial segregation to one of the daughter cells. Left panels: tomographic slices showing the centrosomes (c, with centrioles in purple), the autosomes (a), and the unpaired X chromosome (x) aligned along the spindle axis, mitochondria (m) and fibrous body-membranous organelles (fb). Mid left panels: corresponding three-dimensional models illustrating the organization of the full spindle. Autosomes are in different shades of either blue or cyan, the X chromosome in red, centriolar microtubules in purple, microtubules within 150 nm to the chromosome surfaces in yellow, and all other microtubules in gray. Mid right panels: association of kinetochore microtubules with the kinetochores. Kinetochores are shown as semi-transparent regions around each chromosome. The part of each microtubule entering the kinetochore region around the holocentric chromosomes is shown in green. Right panels: visualization of end-on (white) versus lateral (orange) associations of microtubules with chromosomes. Only the parts of microtubules inside of the kinetochore region are shown. The autosome-to-autosome distance (A-A) for each reconstruction is indicated in the left column. Scale bars, 500 nm.

The online version of this article includes the following video(s) for figure 4:

*Figure 4 continued on next page*

exhibited higher tortuosity, indicating higher curvature (Figures *Appendix 1—figures 6B* and *5C-D*). In addition, laterally associated microtubules had a higher degree of tortuosity compared to end-on associated microtubules. This suggests that other cellular forces, besides those generated by pulling forces, may also be acting on microtubules connected to the lagging X during anaphase I.

## Segregation of the X chromosome correlates with an asymmetry in the number of associated microtubules

To further characterize features of X chromosome lagging and resolution, we determined the ratio of microtubule length in confined volumes on each side of X for each tomographic data set (*Figure 6A*), which were then plotted against autosome-to-autosome distance (*Figure 6B*). We also determined the number of kinetochore microtubules on each side of X and calculated the ratio of these two values (*Figure 6C–D*). The ratio of total microtubule length and microtubule number were about one in metaphase and early anaphase, suggesting microtubules are present equally on both sides. As anaphase progresses, this ratio deviated from one, indicating less microtubules on one side that presumably enable the X to resolve to the opposing side.

Additionally, we tracked microtubules (β-tubulin::GFP) relative to chromosomes (histone H2B::mCherry) by live-cell imaging (*Figure 6E*, upper panel). Measuring the ratio of total GFP fluorescence in similar 3D volumes on each side of the X over time (*Figure 6E*, lower panel) showed a ratio of almost one at early anaphase I, indicating similar microtubule content on each side. As the X chromosome segregated to one side, we detected increased intensity on the side the X moved closer toward (*Figure 6F*). Thus, an asymmetry in associated microtubules correlates with X chromosome resolution, where attachments likely stochastically break as poles separate, allowing the X to resolve to the side with more associated microtubules.

## Interdigitating midzone microtubules are not a prominent feature during spermatocyte anaphase progression

A hallmark of anaphase progression in *C. elegans* embryonic mitosis and oocyte meiosis is a structure of overlapping spindle midzone microtubules that forms between separating chromosomes known as the central spindle (*Yu et al., 2019*). However, the lagging X chromosome in wild-type male spermatocytes precluded detection of such a structure in either light microscopy or electron tomography. We thus examined spermatocytes in *tra-2*(e1094) XX males, where paired X chromosomes do not lag. By live imaging, we detected only a weak microtubule signal in early anaphase I between segregating chromosomes (*Appendix 1—figure 7A*). Interestingly, *tra-2*(e1094) males without a lagging chromosome exhibited a faster spindle elongation rate and a longer final pole-to-pole distance compared to wild-type males that have a lagging X (*Appendix 1—figure 7A, B* and *Figure 1B*). By electron tomography, we detected about 160 microtubules in the anaphase I spindle midzone of *tra-2*(e1094) males (*Appendix 1—figure 7C*; *Appendix 1—figure 7—video 1*). These microtubules, however, did not show an interdigitated pattern characteristic of *C. elegans* mitotic or oocyte meiotic spindle midzones (*Yu et al., 2019*). Thus, even in the absence of a lagging chromosome, male meiotic spindles do not form a typical spindle midzone of overlapping microtubules.

Further, central spindle specifiers that localize in the spindle midzone in oocyte meiosis and mitosis did not localize within the midzone during mid to late sperm meiotic anaphase I in both the presence (*him-8*(e1489)) or absence (*tra-2*(e1094)) of a lagging X chromosome (*Appendix 1—figure 8A*).

**Table 1.** Analysis of tomographic data sets used throughout this study.

| Spindle parameters | Full data sets | | | | | | | | Partial data sets | | |
|---|---|---|---|---|---|---|---|---|---|---|---|
| | Metaphase no. 1 | Metaphase no. 2 | Metaphase no. 3 | Anaphase onset | Anaphase no. 1 | Anaphase no. 3 | Anaphase no. 4 | Anaphase no. 7 | Anaphase no. 2 | Anaphase no. 5 | Anaphase no. 6 |
| MTs total | 1729 | 2406 | 1689 | 2051 | 1405 | 1540 | 1403 | 1881 | (893) | (671) | (246) |
| MTs within 150 nm from chromosomes (KMTs) | 524 | 912 | 650 | 794 | 633 | 821 | 752 | 944 | (580) | (499) | (160) |
| End-on associated KMTs on X-chromosome | 29 | 38 | 38 | 53 | 27 | 50 | 38 | 42 | 57 | 30 | 43 |
| Lateral associated KMTs on X-chromosome | 79 | 34 | 91 | 22 | 61 | 55 | 33 | 34 | 47 | 52 | 28 |
| End-on associated KMTs on autosomes | 154 | 355 | 199 | 318 | 106 | 189 | 181 | 175 | | | |
| Lateral associated KMTs on autosomes | 262 | 485 | 321 | 400 | 437 | 527 | 500 | 692 | | | |
| Autosome-to-autosome distance [μm] | 0.76 | 0.94 | 0.99 | 1.00 | 2.98 | 3.35 | 3.37 | 5.44 | (3.14) | (3.58) | (4.47) |
| Autosomes1-to-X distance [μm] | 0.37 | 0.43 | 0.43 | 0.47 | 1.45 | 1.45 | 1.37 | 1.95 | (1.54) | (1.71) | |
| Autosomes2-to-X distance [μm] | 0.39 | 0.51 | 0.56 | 0.54 | 1.56 | 2.02 | 1.99 | 3.51 | (1.72) | (1.90) | |
| Pole-to-pole distance [μm] | 3.10 | 3.41 | 3.45 | 3.51 | 4.97 | 5.22 | 4.99 | 7.04 | | | |
| Pole1-to-X distance [μm] | 1.64 | 1.66 | 1.66 | 1.62 | 2.60 | 2.29 | 2.21 | 2.69 | | | |
| Pole2-to-X distance [μm] | 1.48 | 1.76 | 1.80 | 1.90 | 2.39 | 3.04 | 2.77 | 4.38 | | | |
| Autosome-to-centrosome distance [μm] | 1.18 | 1.24 | 1.23 | 1.26 | 1.00 | 0.94 | 0.82 | 0.83 | | | |
| Mother-to-daughter centriole distance [μm] | 0.26 | 0.21 | 0.27 | 0.35 | 0.37 | 0.73 | 0.74 | 1.11 | | | |
| Original name of data set | T0391_worm13 metaphase01 | T0391_worm14 metaphase | T0391_worm13 metaphase02 | T0391_worm13 meta-anaphase01 | T0391_worm05 anaphase02 | T0391_worm08 lateanaphase | T0391_anaphase01 early | T0391_worm09 late_anaphase | T0391_worm07b | T0391_worm06 | T0391_worm02 |
| Number of sections | 14 | 11 | 14 | 25 | 14 | 17 | 11 | 30 | 8 | 4 | 6 |
| Est. tomographic volume [μm$^3$] | 102.33 | 94.70 | 107.52 | 115.12 | 113.96 | 131.92 | 101.51 | 268.31 | 36.99 | 18.52 | 30.37 |

The table summarizes all microtubule numbers and distances as measured within the electron tomographic reconstructions in this study. A kinetochore microtubule (KMT) is defined as a microtubule that is at least 150 nm from the surface of a chromosome. KMTs are sub-divided into end-on and lateral associated MTs. End-on KMTs are defined as pointing towards the chromosome surface, lateral MTs are all remaining KMTs. Distances were measured between the geometric centers of autosomes (mean position of individual autosomes), centrosomes (center point of both centrioles) and centrioles (between the centers of the mother and daughter centriole). Tomographic volumes were estimated by multiplying the X-Y dimensions of each tomogram with the number of sections (with a section thickness of 300 nm).

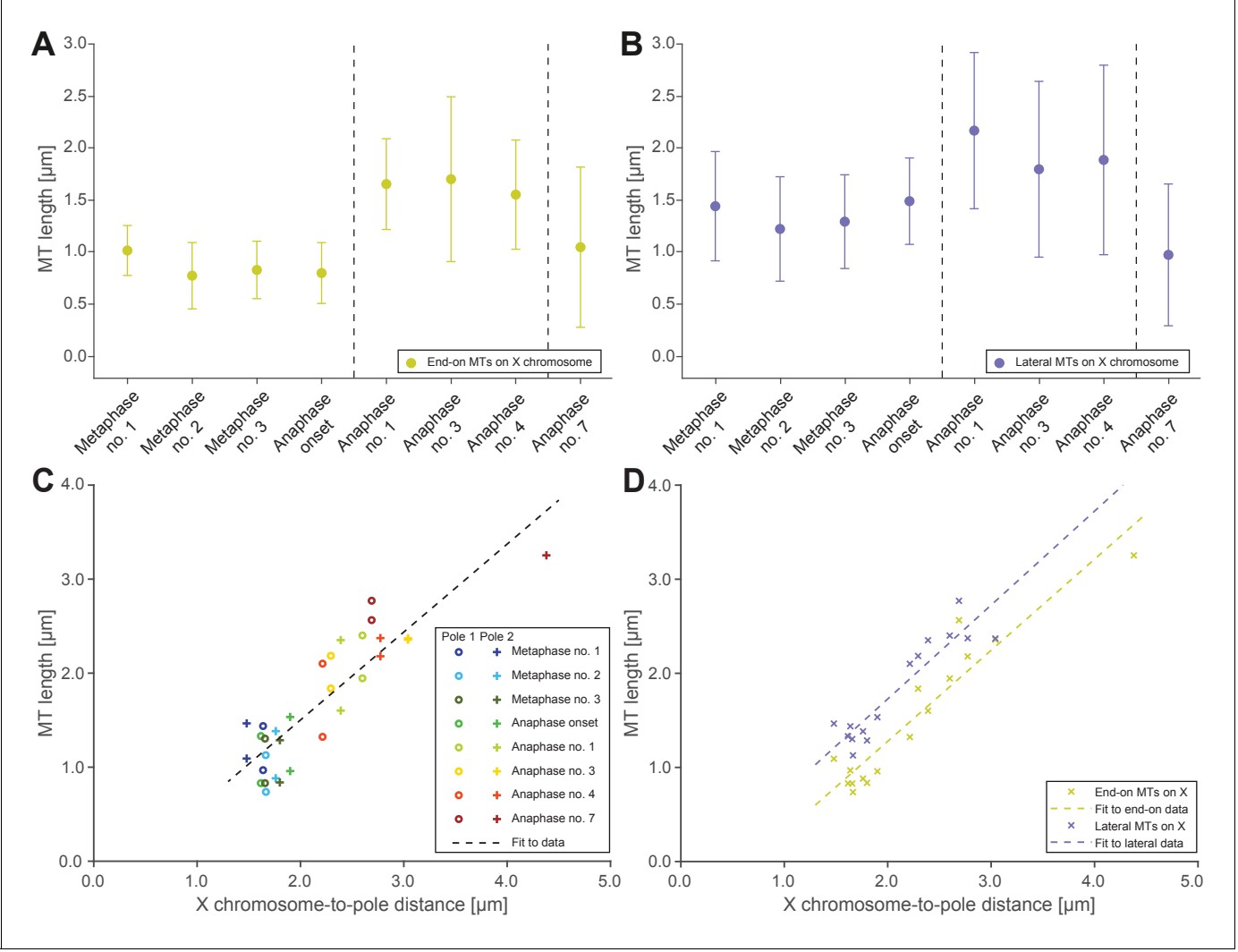

**Figure 5.** Microtubules that connect the X chromosome to centrosomes are continuous and lengthen during anaphase I. (A) Length distribution of end-on X chromosome-associated microtubules at different stages of meiosis I. Dots show the mean, error bars indicate the standard deviation. Dashed lines indicate a grouping of the spindles according to the meiotic stages: metaphase/anaphase onset, mid anaphase and late anaphase (see also *Appendix 1—figure 2*). (B) Length distribution of lateral X chromosome-associated microtubules at different stages of meiosis I. (C) Mean length of microtubules plotted for each side of the X chromosome against the respective X chromosome-to-pole distance (each tomographic data set color-coded). The values for end-on and laterally associated microtubules are given separately. The measurements were performed on all data sets as shown in *Figures 3* and *4*. A trend line was fitted to indicate the linear relationship between microtubule length and chromosome-to-pole distance. (D) Similar plot as in (C) but end-on (yellow) and laterally (purple) X chromosome-associated microtubules are shown. Two trend lines were fitted to the data sets to illustrate linear relationships independent of the type of association of the microtubules with the X chromosome.

The online version of this article includes the following source data for figure 5:

**Source data 1.** Measurements of microtubule lengths in spindles shown in *Figures 3* and *4* used to generate data in *Figure 5*.

First, Aurora B[AIR-2], a component of the chromosomal passenger complex (*Davies et al., 2014*; *de Carvalho et al., 2008*; *Dumont et al., 2010*; *Maton et al., 2015*; *Schumacher et al., 1998*; *Severson et al., 2000*), associated with separating autosomes or the lagging X during anaphase I. CLASP[CLS-2], a microtubule stabilizer (*Dumont et al., 2010*; *Maton et al., 2015*; *Nahaboo et al., 2015*), localized to the inside face of separating chromosomes and remained chromosome-associated during anaphase I. A centralspindlin component, MKLP1[ZEN-4], localized between separating chromosomes at very early anaphase, then to the cell membrane in a ring-like structure likely on the ingressing furrow at mid-anaphase (*Powers et al., 1998*; *Raich et al., 1998*). Similarly, using live-imaging, PRC1[SPD-1] (*Mullen and Wignall, 2017*; *Nahaboo et al., 2015*; *Verbrugghe and White*,

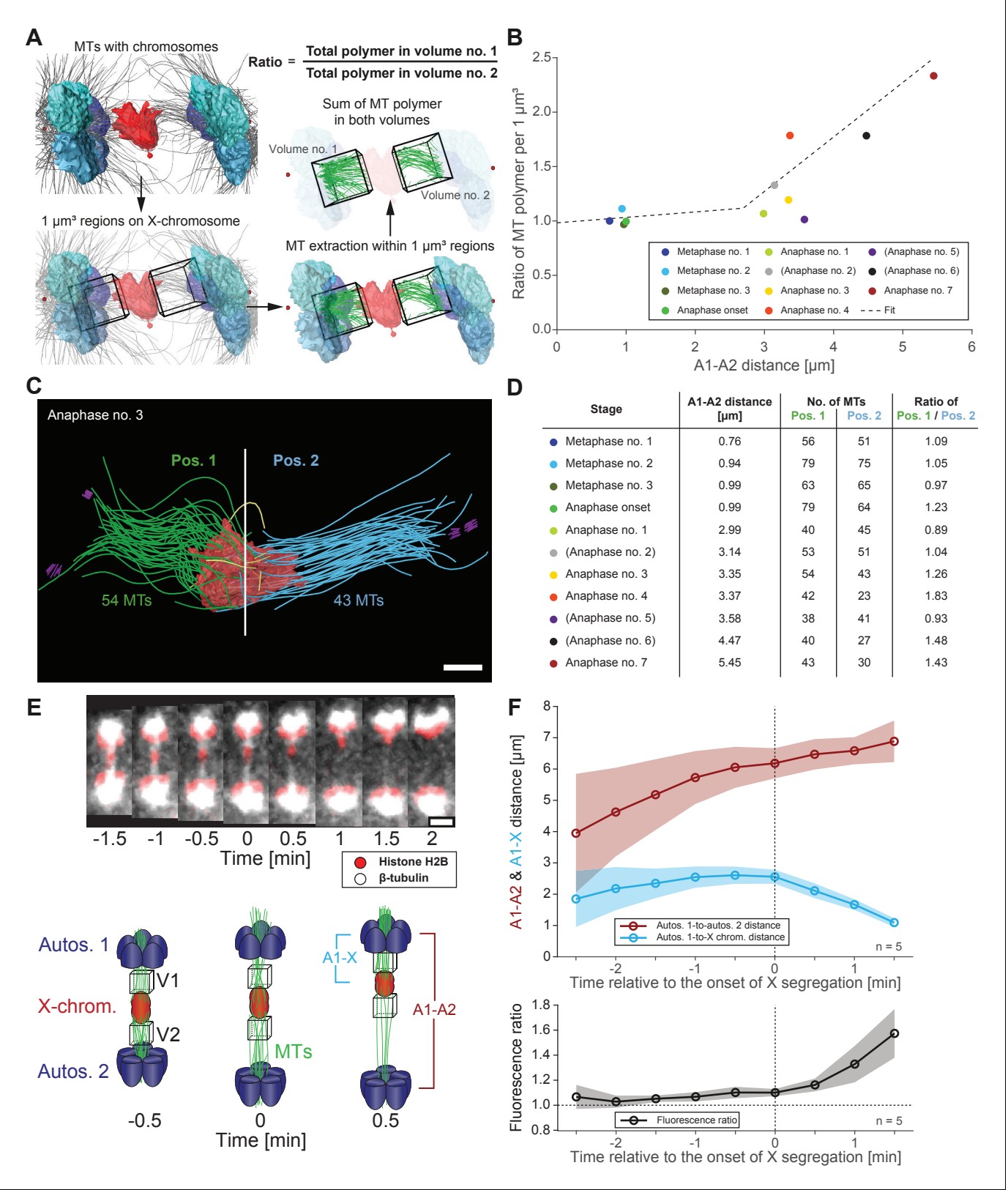

**Figure 6.** Resolution of the X chromosome to one side correlates with an asymmetry of microtubules. (**A**) Left: three-dimensional model of an anaphase I spindle and definition of two equal volumes on opposite sides of the X chromosome. Right: Measurement of total polymer length within selected volumes of 1 μm³. (**B**) Graph showing the ratio of both volumetric measurements plotted against the autosome-to-autosome distance for the meiotic

*Figure 6 continued on next page*

Figure 6 continued

stages as shown in *Figures 3* and *4* and *Appendix 1—figure 4*. A trend line was fitted to illustrate the increase in the asymmetry. (C) Deconstructed 3D model (data set anaphase no. 3) illustrating the microtubules associated with each side of the X chromosome (red), named pos. 1 (green) and pos. 2 (blue). Centrioles are shown in purple, microtubules not connected to the spindle poles in yellow. Scale bar, 500 nm. (D) Table showing the autosome 1-to-autosome 2 distance (A1, A2), the total number of microtubules for both positions and the calculated ratio for each data set (see *Figures 3* and *4* and *Appendix 1—figure 4* (data sets shown in parentheses)). (E) Upper panel: Maximum intensity projection images from live imaging showing microtubules labeled with β-tubulin::GFP (white) and chromosomes with histone H2B::mCherry (red). Time is given relative to the onset of X chromosome segregation (t = 0). Scale bar, 2 μm. Lower panel: Illustration of the measurement of fluorescence intensity in two volumes (V1, V2) of 1 μm$^3$ at opposite sides of the X chromosome (red). Autosomes are in blue, microtubules in green. (F) Ratio of fluorescence intensities as measured in (E). Upper panel: autosomes 1-to-autosomes two distance (A1-A2, red) and autosomes 1-to-X chromosome distance (A1-X, blue) over time. Solid lines show the mean, shaded areas indicate the standard deviation. The onset of X chromosome movement is given as time point zero (t = 0). Lower panel: ratio of fluorescence intensities (V1/V2) for corresponding time points (black, time is relative to the onset of segregation of the X chromosome, t = 0; n = 5).

The online version of this article includes the following source data for figure 6:

**Source data 1.** Autosome-to-autosome distances and volumetric measurements for the meiotic stages shown in *Figures 3* and *4* and *Appendix 1—figure 4* used to generate *Figure 6B*.

*2004*), a known microtubule bundling factor, initially briefly localized between segregating autosomes and around the X chromosome in very early anaphase I, but rapidly disappeared as anaphase I progressed in males and hermaphrodite spermatocytes (*Appendix 1—figure 8B*). Thus, spindle elongation, the segregation of autosomes, and the resolution of the X continued even without a detectable signal of PRC1$^{SPD-1}$ in between chromosomes. Overall, these results suggest that male meiotic spindles, in the presence or absence of a lagging X, do not form a 'canonical' midzone structure during mid-to-late anaphase I.

## Spermatocyte meiotic spindles display both anaphase A and anaphase B movement

We next investigated how spermatocyte meiotic spindles drive chromosome movement over time. We measured changes in pole-to-pole (P-P), autosome-to-autosome (A-A) and pole-to-autosome (P-A) distances during both meiotic divisions using a strain with centrosomes labeled with γ-tubulin::GFP and chromosomes with H2B::mCherry. In meiosis I (*Figure 7A–B*; *Figure 7—video 1*), the pole-to-pole distance increased from 4.1 ± 0.3 μm to 8.0 ± 0.6 μm (mean ± SD; n = 31) with an elongation speed of 1.29 ± 0.36 μm/min (*Figure 7E*). This speed is significantly higher compared to 0.6–0.8 μm/min reported for both female meiotic divisions (*McNally et al., 2016*). We also found a simultaneous anaphase A-type movement in spermatocytes with pole-to-autosome distance decreased by half, from 1.6 ± 0.3 μm to 0.8 ± 0.3 μm and a speed of 0.39 ± 0.27 μm/min (*Table 2*). In addition, the autosome-to-autosome distance increased from 0.9 ± 0.2 μm to 6.5 ± 0.4 μm (mean ± SD; n = 31). Roughly, from this 5 μm increase in autosome-to-autosome distance, anaphase B provides about 4 μm (~80%) of separation, whereas anaphase A provides only 1 μm (~20%). Chromosome dynamics in meiosis II also exhibited anaphase A and anaphase B-type movements (*Figure 7C–D and F*; *Figure 7—video 2*; *Table 2*).

Taken together, spermatocyte meiotic spindles in *C. elegans* exhibit both anaphase A and B-type movements. This is distinct from mitosis in the early *C. elegans* embryo, which utilizes only anaphase B mechanisms (*Oegema et al., 2001*), or oocyte meiosis, which uses acentrosomal mechanisms (*Dumont et al., 2010*; *McNally et al., 2016*; *Muscat et al., 2015*; *Redemann et al., 2018*). Similar to grasshopper spermatocytes (*Ris, 1949*), anaphase A and B movement occurs simultaneously, with anaphase A contributing approximately one fifth to the overall chromosome displacement.

## Electron tomography does not suggest a shortening of autosome-associated kinetochore microtubules during anaphase

A well-described mechanism for anaphase A (i.e. a decrease in chromosome-to-pole length) is microtubule shortening (*Asbury, 2017*). We thus analyzed individual kinetochore microtubule lengths in our 3D EM reconstructions. We set the metaphase I (data set Metaphase no. 1) as the earliest, since end-on kinetochore microtubules associated to autosomes were slightly longer than all other data sets (0.79 ± 0.3 μm; n = 153; *Figure 7G*). We speculate at this point chromosomes may not be fully

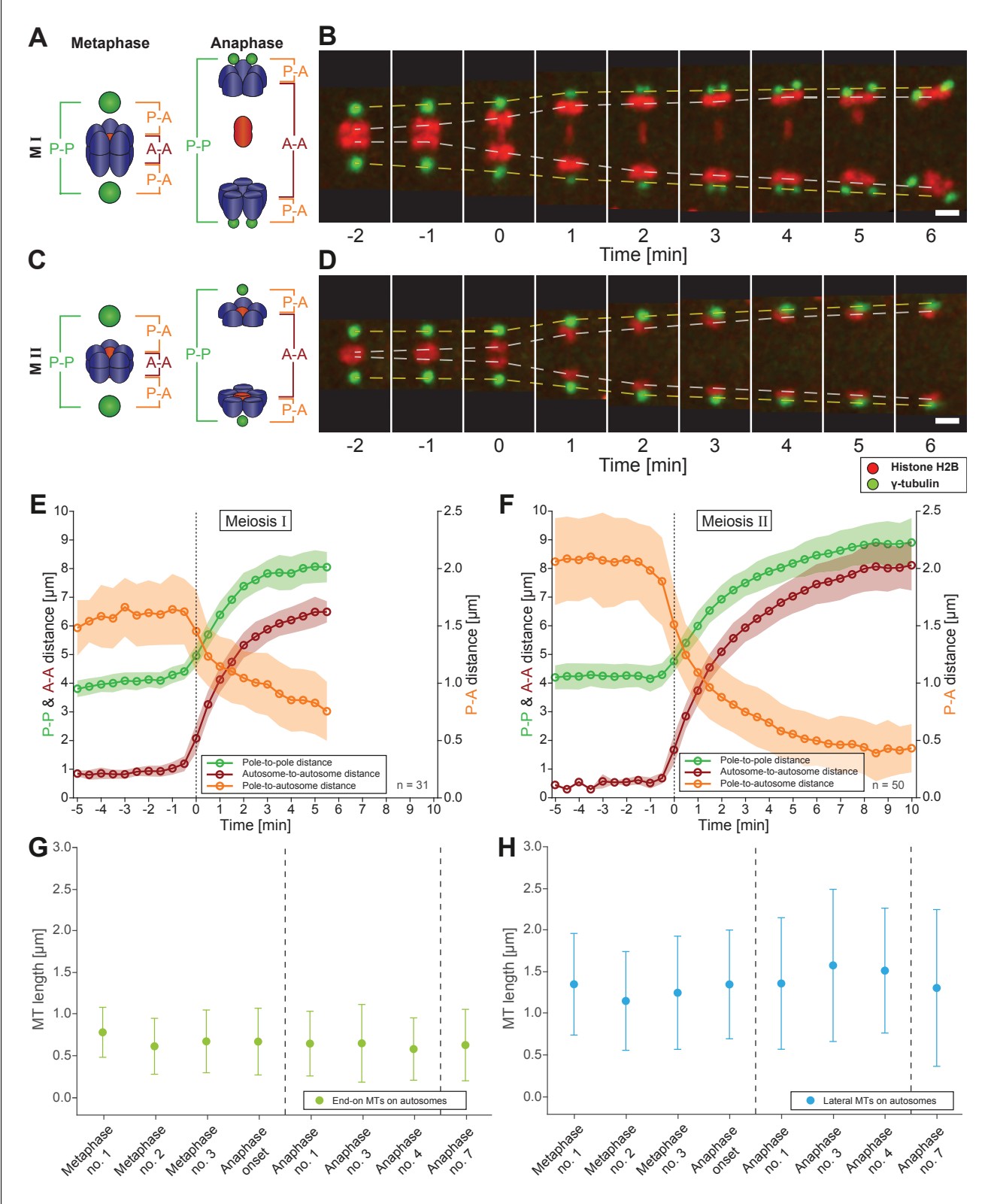

**Figure 7.** Spermatocyte meiotic spindles display both anaphase A and B movement. (**A**) Schematic representation of metaphase and anaphase during meiosis I. Centrosomes are in green, autosomes in blue, and the univalent X chromosome in red. The pole-to-pole (P-P, green), autosome-to-autosome (A-A, red), and both pole-to-autosome distances (P-A, orange) are indicated. (**B**) Time series of confocal image projections of a spindle in meiosis I with centrosomes labeled with γ-tubulin::GFP (green) and chromosomes with histone H2B::mCherry (red). The separation of the centrosomes (yellow dashed

*Figure 7 continued on next page*

*Figure 7 continued*

line) and the autosomes (white dashed line) over time is indicated. Anaphase onset is time point zero (t = 0). Scale bar, 2 µm. (C) Schematic representation of metaphase and anaphase during meiosis II. (D) Separation of centrosomes and autosomes in meiosis II as in (F). Scale bar, 2 µm. (E) Quantitative analysis of autosome and centrosome dynamics in meiosis I show a decrease in pole-autosome distance that is characteristic of anaphase A. Anaphase onset is time point zero (t = 0). The mean and standard deviation is given (circles and shaded areas). (F) Quantitative analysis of autosome and centrosome dynamics in meiosis II. (G) Length distribution of end-on autosome-associated kinetochore microtubules at different stages of meiosis I. Dots show the mean, error bars indicate the standard deviation. Dashed lines indicate a grouping of the spindles according to the meiotic stages: metaphase/anaphase onset, mid anaphase and late anaphase (see also *Appendix 1—figure 2*). (H) Length distribution of laterally autosome-associated kinetochore microtubules at different stages of meiosis I.

The online version of this article includes the following video and source data for figure 7:

**Source data 1.** Measurements of autosome and centrosome dynamics from replicates used in *Figure 7* .
**Figure 7—video 1.** Spindle dynamics in wild-type spermatocytes in meiosis I.
https://elifesciences.org/articles/50988#fig7video1
**Figure 7—video 2.** Spindle dynamics in wild-type spermatocyte meiosis II.
https://elifesciences.org/articles/50988#fig7video2

under tension. In both other metaphase data sets, the length of end-on associated microtubules was 0.62 and 0.68 µm. Unexpectedly, as anaphase I progressed and the autosome-to-pole distance decreased, we observed that end-on kinetochore microtubules did not significantly shorten, remaining at 0.59–0.65 µm (*Appendix 1—figure 9A* and *Appendix 1—figure 5E*). Interestingly, the length of laterally associated microtubules did increase from 1.28 µm in metaphase to 1.44 µm in anaphase (*Figure 7H*, *Appendix 1—figures 9B* and *5F*). Thus, unlike in other systems (*Asbury, 2017*), our tomographic analysis suggests that shortening of kinetochore microtubules does not fully account for the anaphase A observed by light microscopy.

## Tension release across the spindle may contribute to autosomal anaphase A

To account for anaphase A in spermatocyte meiosis, we hypothesized that changes in the shape of chromosomes, centrosomes, and the association angles of kinetochore microtubules with autosomes induced by tension released at the metaphase to anaphase transition (*Dumont and Mitchison, 2009*; *Gardner et al., 2005*) may contribute to the decrease in chromosome-to-centrosome distance.

First, to examine the release of chromosome stretch that peaked at metaphase chromosome alignment, we measured individual autosome expansion along the spindle axis by plotting the cross-sectional areas over the chromosome distance. This generated a stretch value obtained at the Full Width at Half-Maximum (FWHM) of a Gaussian fit to the cross-sectional area along the spindle axis (*Figure 8A*; see Materials and methods). The autosomes of the first metaphase data set were the least stretched (0.52 ± 0.03 µm), consistent that the chromosomes in this data set were not yet under full tension. Chromosomes in metaphase data set no. two were most stretched (0.73 ± 0.12 µm). As chromosomes separated, autosomes rounded up to a value of 0.56 µm in anaphase data sets no. four and no. 7 (*Figure 8B*, *Appendix 1—figure 5G*). This is about 23% less compared to metaphase no. 2, thereby moving chromosome centers closer to the poles. Thus, the release during anaphase of chromosome stretch induced by metaphase alignment accounts for a portion of anaphase A pole-chromosome shortening.

Second, we considered that as centrosomes split and shift from a spherical to a stretched shape, spindle poles may thus move closer to chromosomes (*Appendix 1—figure 3*). Because the outline of centrosomes cannot be clearly distinguished in EM data, we measured the distance of the plus-end of the kinetochore microtubules to the closest centriole (*Figure 8C*). This distance significantly shortened when comparing metaphase data sets (0.99–1.14 µm) to the anaphase data sets (0.78–0.95 µm; *Figure 8D*, *Appendix 1—figure 5H*), resulting in autosomes being 0.2 µm or 20% closer to centrioles.

Third, we hypothesized tension release would also alter the attachment angle of end-on kinetochore microtubules with autosomes, bringing chromosomes closer to spindle poles. We determined the angle between each kinetochore microtubule plus-end at the chromosome surface and each centrosome-chromosome axis (*Figure 8E*). The angle in the metaphase data sets was 37˚- 41˚. As

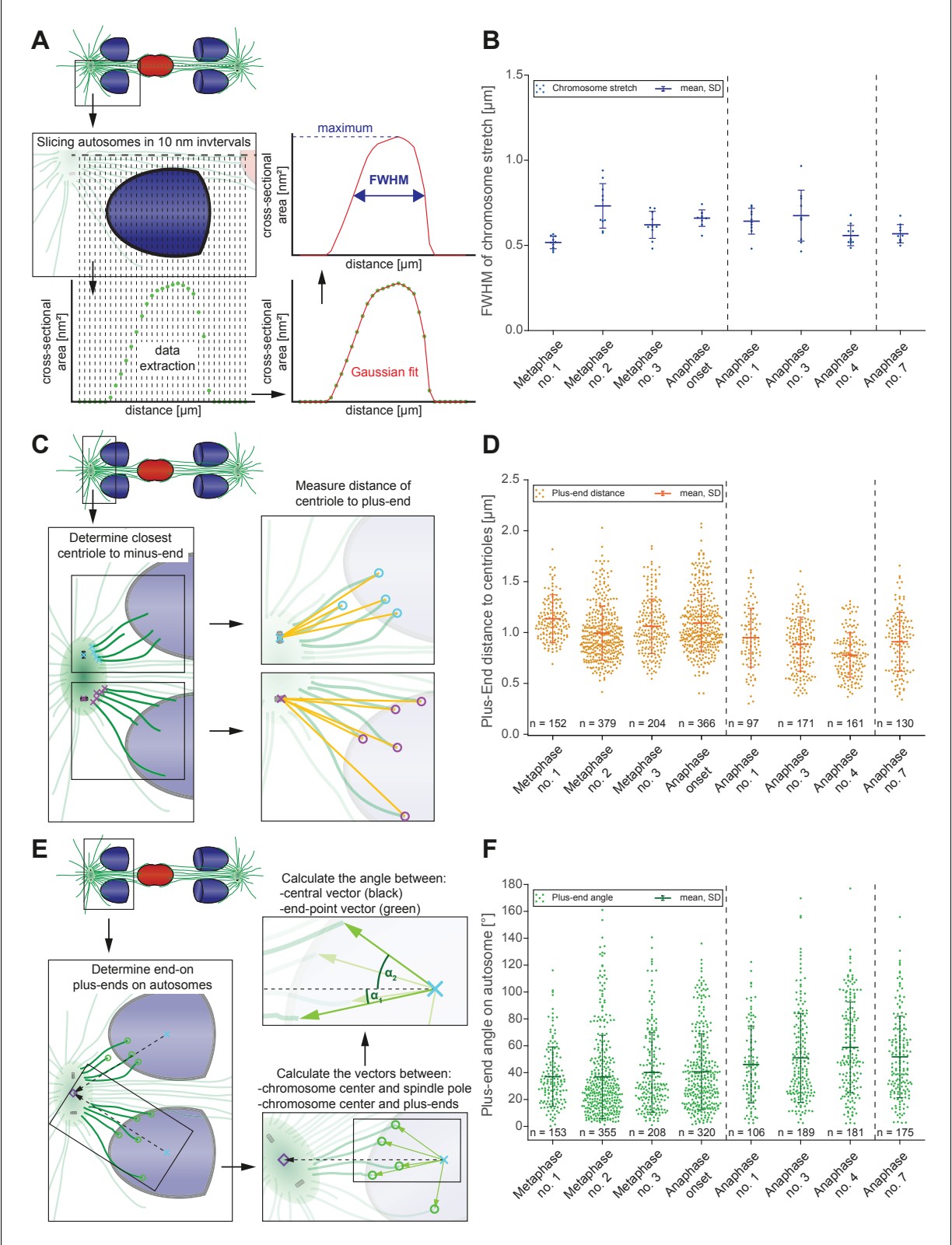

**Figure 8.** Changes in spindle geometry during anaphase A. (**A**) Analysis of autosome stretching in anaphase I. Schematic representation showing how the full width half maximum (FWHM) of stretching along the pole-to-pole axis for a single autosome is assessed (see also Materials and methods). (**B**) FWHM of chromosome stretch for all autosomes at each meiotic stage shown in *Figures 3* and *4*. The mean, the standard deviation and the number of measurements (n = 10) for each meiotic stage are given. Dashed lines indicate a grouping of the spindles according to the meiotic stages: metaphase/

*Figure 8 continued on next page*

*Figure 8 continued*

anaphase onset, mid anaphase and late anaphase (see also *Appendix 1—figure 2*). Additional ANOVA results are shown in *Appendix 1—figure 5G*. (C) Schematic depicting the determination of the distance of individual kinetochore microtubule plus ends to the closest centriole. For each kinetochore microtubule (green line), the direct distance (yellow line) from the putative plus end to the respective centriole was measured (plus ends of kinetochore microtubules are shown in circles). (D) Distance of kinetochore microtubule plus-ends to centrioles for each meiotic stage. Additional ANOVA results are shown in *Appendix 1—figure 5H*. (E) Analysis of the attachment angle of end-on-associated autosomal kinetochore microtubules. The schematic illustrates the defined main axis for the measurements (dashed line from the center of each autosome to the center of the centrosome). The angle (α) between each line connecting the kinetochore microtubule plus-end and the autosome center (green lines) and the main axis (dashed line) was measured for each kinetochore microtubule. (F) Plot showing the angle measurements for all data sets. Additional ANOVA results are given in *Appendix 1—figure 5I*.

The online version of this article includes the following source data for figure 8:

**Source data 1.** Measurements for Full Width at Half-Maximum(FWHM) of replicates in each stage shown in *Figures 3* and *4* used in *Figure 8*.

autosomes rounded up during anaphase, the attachment angle in the anaphase data sets increased to 46˚- 59˚, bringing chromosomes closer to poles (*Figure 8F*, *Appendix 1—figure 5I*). Simple trigonometric calculations with a constant microtubule length of 0.63 µm found this increase in the attachment angle contributes about 0.17 µm or 17% shortening in chromosome-to-pole distance.

In sum, we identify three factors that contribute to pole-chromosomes shortening during anaphase: 1) a loss of chromosome stretch during anaphase, which shortens the chromosome by about 0.34 µm; 2) changes in centrosome size and shape that contributes about 0.2 µm; and 3) the opening of the attachment angle that accounts for 0.17 µm. These factors comprise ~70% of the total ~1 µm chromosome-to-pole distance shortening observed in spermatocyte meiosis, though this may be underestimated due to limitations in the tomographic reconstruction from serial semi-thick sections. Overall, our ultrastructure analysis revealed previously unknown, alternative mechanisms that contribute to anaphase-A movement.

## Discussion

Prior to this work, only a handful of studies addressed spermatocyte spindle dynamics during male meiosis (*Fegaras and Forer, 2018*; *Felt et al., 2017*; *Golding and Paliulis, 2011*; *LaFountain et al., 2011*; *LaFountain et al., 2012*; *Nicklas and Kubai, 1985*; *Nicklas et al., 2001*; *Zhang and Nicklas, 1995*). Our large-scale tomographic reconstruction combined with live imaging and immunostaining

**Table 2.** Measurements of spindle dynamics in male meiosis.

| Distance | Spindle parameter | Meiosis I | | Meiosis II | |
|---|---|---|---|---|---|
| | | Mean | SD | Mean | SD |
| P-P[1] | Initial spindle length (metaphase) | 4.1 µm | ±0.3 µm | 4.2 µm | ±0.4 µm |
| | Final spindle length (end of anaphase) | 8.0 µm | ±0.6 µm | 8.8 µm | ±0.8 µm |
| | Initial rate (1 st minute) | 1.29 µm/min | ±0.36 µm/min | 1.11 µm/min | ±0.42 µm/min |
| | Duration of elongation | 3–4 min | | 8–9 min | |
| A-A[2] | Initial spindle length (metaphase) | 0.9 µm | ±0.2 µm | 0.6 µm | ±0.2 µm |
| | Final spindle length (end of anaphase) | 6.5 µm | ±0.4 µm | 8.0 µm | ±0.8 µm |
| | Initial rate (1 st minute) | 2.07 µm/min | ±0.37 µm/min | 2.16 µm/min | ±0.32 µm/min |
| | Duration of elongation | 4–5 min | | 8–9 min | |
| P-A[3] | Initial spindle length (metaphase) | 1.6 µm | ±0.3 µm | 2.0 µm | ±0.4 µm |
| | Final spindle length (end of anaphase) | 0.8 µm | ±0.3 µm | 0.4 µm | ±0.2 µm |
| | Initial rate (1 st minute) | −0.39 µm/min | ±0.27 µm/min | −0.64 µm/min | ±0.33 µm/min |

Distances: [1]P-P, pole-to-pole distance; [2]A-A, autosome-to-autosome distance; [3]P-A, pole-to-autosome distance. Initial spindle length is given at metaphase, final distance refers to the end of anaphase when spindle elongation plateaus. Values are given as mean values (± standard deviation, SD). The numbers of analyzed spindles are: n = 31 for meiosis I; n = 50 for meiosis II.

in *C. elegans* now provides an in-depth characterization of the molecular architecture and dynamics of the male meiotic spindle.

## Lagging and resolution of the X chromosome

We determined new molecular features of chromosome lagging and resolution in *C. elegans*. We show the absence of a pairing partner and/or the inability to pair induces any chromosome to lag. Furthermore, microtubule 'bridges' to lagging chromosomes consist of continuous microtubules that attach to each side and lengthen during anaphase I. Moreover, we detected both end-on and laterally-associated microtubules on the lagging X. This is distinct from embryonic mitosis, where the vast majority of microtubules make end-on attachments to rod-like chromosomes (*O'Toole et al., 2003*; *Redemann et al., 2017*). How is lengthening of X-associated microtubules achieved? Possibly, kinetochore microtubules grow at their plus ends as poles move apart. Microtubule growth at a similar rate to spindle elongation would maintain kinetochore microtubule association to the X. Alternatively, the growth rate of kinetochore microtubules could exceed the rate of spindle elongation, thus allowing minus-end directed interactions of motor proteins such as dynein (*Reck-Peterson et al., 2018*; *Schmidt et al., 2005*; *Schmidt et al., 2017*). Possible roles for dynein in lagging chromosome segregation, the function of end-on *versus* lateral associations of microtubules to the lagging X, and the influence of the meiotic kinetochore shape and connections can now be addressed in future studies in *C. elegans* and other systems to further understand lagging chromosome generation and resolution.

A crucial question is how lagging chromosomes resolve during anaphase I. We find the lagging X chromosome is subject to pulling forces mediated by microtubules as poles move apart (*Figure 2*). Our analyses support that an imbalance of pulling forces may result stochastically from continuous attachment and detachment of kinetochore microtubules. In such a 'tug-of-war' model, the side that maintains more connections wins (*Appendix 1—figure 10A*). A similar mechanism has also been suggested during chromosomal oscillations at mitotic prometaphase and metaphase (*Ault et al., 1991*; *Skibbens et al., 1993*; *Soppina et al., 2009*) with chromokinesins and dynein as possible candidates for switching the direction of the oscillations (*Sutradhar and Paul, 2014*). As for the initiation of segregation, an analogous situation is the segregation of merotelically attached mammalian kinetochores, where microtubule breakage was suggested to initiate the segregation of lagging mitotic chromosomes (*Cimini et al., 2004*). Additional tomographic analysis of spindles at late stages of anaphase I will be key to further support our proposed model of lagging chromosome resolution.

Importantly, many species have evolved distinct spindle structures and segregation strategies to resolve lagging of sex or unequal numbers of chromosomes (*Fabig et al., 2016*; *Shakes et al., 2011*; *Winter et al., 2017*). Segregation in cells with aneuploidy and chromosomal abnormalities are potential drivers of infertility (*Barri et al., 2005*; *García-Mengual et al., 2019*; *Hassold and Hunt, 2001*; *Ioannou and Tempest, 2015*) and cancer progression (*Bolhaqueiro et al., 2019*; *Chunduri and Storchová, 2019*; *Ly et al., 2019*). Thus, our studies can impact the understanding of partition mechanisms in other systems that segregate both paired and lagging chromosomes to efficiently and reliably generate cells with correct ploidy (*Fabig et al., 2016*).

## Contributors of anaphase A not reliant on shortening of kinetochore microtubules

The single-microtubule resolution of electron tomography unexpectedly revealed the lengths of autosomal end-on kinetochore microtubules are largely constant during sperm anaphase. This is in contrast to the kinetochore microtubule shortening typically associated with anaphase A observed by light microscopy in many systems (*Asbury, 2017*). Further, we developed methods to use ultrastructural data to identify three contributors to spermatocyte anaphase A (*Figure 8*). First, autosomes stretched at metaphase relax from tension released by separase-mediated cleavage of cohesins at anaphase (*Severson and Meyer, 2014*), resulting in chromosome shape change. Second, spindle poles decrease in size and change shape as centrioles split, which also shortens distance between microtubule plus-ends and centrioles. Third, the ends of microtubules on chromosomes shift from a central to more peripheral position during anaphase, decreasing the microtubule-to-pole distance. All three factors change the relative position of centrosomes, autosomes and

kinetochore microtubules to one another independent of kinetochore microtubule shortening. Microtubules can shorten during anaphase I, as observed when the X resolves to one side; thus, a shortening of a subset of microtubules that is difficult to detect by current methods may also contribute to a small portion of anaphase A movement. Nonetheless, our proposed new mechanisms can now be considered when analyzing anaphase A movement in other systems (*Appendix 1—figure 10B*).

## Distinctions of *C. elegans* spindles in spermatocytes

In *C. elegans*, we found important differences in the molecular composition of meiotic spindles in spermatocytes compared to those in oocyte meiosis and embryonic mitosis. First, outer kinetochore proteins are retained and microtubules remain associated to chromosomes between meiotic divisions in sharp contrast to oocyte meiotic anaphase, where kinetochore levels diminish dramatically and microtubules disassemble during anaphase I progression (*Dumont et al., 2010*).

Second, autosome-associated kinetochore microtubules are continuous and directly attached to poles, in contrast to embryonic mitosis, where chromosome-connected microtubules end in the spindle matrix and make indirect contact with microtubules attached to poles (*Redemann et al., 2017*). We speculate that the direct kinetochore-to-pole connection in spermatocytes is related to the small size of spermatocytes, in contrast to the relatively large one-cell embryo (*Redemann et al., 2017*).

Third, we find sperm-specific differences in central spindle architecture during anaphase I that also may impact the lagging and resolution of the X chromosome. When the X lags, microtubule bridges connected to X dominate the spindle midzone. (*Figures 1A* and *4*). With no lagging chromosome in meiosis I in *tra-2*(e1090) mutant males, few microtubules are found between the separated chromosomes (*Figure 1B* and *Appendix 1—figure 7*) and they lack the classical overlapping structure thought to push chromosomes apart, as observed in either acentrosomal oocyte meiosis (*Dumont et al., 2010*; *Laband et al., 2017*; *Redemann et al., 2018*; *Yu et al., 2019*) or centrosomal embryonic mitosis (*Nahaboo et al., 2015*; *Yu et al., 2019*). Furthermore, central spindle specifiers have sperm-specific localization patterns at mid-late anaphase I instead of residing strongly within the spindle midzone (*Davies et al., 2014*; *de Carvalho et al., 2008*; *Dumont et al., 2010*; *Maton et al., 2015*; *Schumacher et al., 1998*; *Severson et al., 2000*). Aurora B Kinase$^{AIR-2}$ and CLASP$^{CLS-2}$ stay associated with spermatocyte chromosomes (*Appendix 1—figure 8A*). Likewise, MKLP1$^{ZEN-4}$, a centralspindlin component (*Powers et al., 1998*; *Raich et al., 1998*), localizes at the ingressing furrow. PRC1$^{SPD-1}$, a microtubule bundling protein (*Verbrugghe and White, 2004*), while transiently present in very early anaphase becomes undetectable (*Appendix 1—figure 8B*). Thus, our results support that *C. elegans* spermatocyte meiosis forms an alternate spindle midzone structure compared with mitosis or oocyte meiosis. We speculate this may have evolved to aid lagging chromosome resolution, and it will be interesting to analyze meiotic spindle midzones in other systems with lagging sex chromosomes (*Fabig et al., 2016*). Alternatively, this midzone organization may stem from mechanisms that regulate cytokinesis to specify intracellular bridges that maintain the germline syncytium (*Lee et al., 2018*; *Zhou et al., 2013*).

*In toto*, our approach combining quantification of 3D ultrastructure of staged spindles with live imaging and immunostaining in males to identify sperm-specific features of meiosis lays the groundwork for further detailed studies on chromosome segregation and provides the necessary analytical tools for analyses on spindles in a broad range of different contexts.

## Materials and methods

**Key resources table**

| Reagent type (species) or resource | Designation | Source or reference | Identifiers | Additional information |
|---|---|---|---|---|
| Genetic reagent *C. elegans* | N2 | (*Brenner, 1974*) | | |
| Genetic reagent *C. elegans* | ANA0072 | (*Nahaboo et al., 2015*) | | |

*Continued on next page*

*Continued*

| Reagent type (species) or resource | Designation | Source or reference | Identifiers | Additional information |
|---|---|---|---|---|
| Genetic reagent *C. elegans* | CB1489 | (*Herman and Kari, 1989*; *Phillips et al., 2005*) | | |
| Genetic reagent *C. elegans* | CB2580 | (*Hodgkin, 1985*) | | |
| Genetic reagent *C. elegans* | MAS91 | (*Han et al., 2015*) | | |
| Genetic reagent *C. elegans* | MAS96 | M. Srayko, Alberta | | Strain maintained in the Srayko lab |
| Genetic reagent *C. elegans* | TMR17 | this study | | Strain maintained in the Müller-Reichert lab |
| Genetic reagent *C. elegans* | TMR18 | this study | | Strain maintained in the Müller-Reichert lab |
| Genetic reagent *C. elegans* | TMR26 | this study | | Strain maintained in the Müller-Reichert lab |
| Genetic reagent *C. elegans* | XC110 | this study | | Strain maintained in the Chu lab |
| Genetic reagent *C. elegans* | XC116 | this study | | Strain maintained in the Chu lab |
| Genetic reagent *C. elegans* | SP346 | (*Madl and Herman, 1979*) | | |
| Genetic reagent *E. coli* | OP50 | (*Brenner, 1974*) | | |
| Antibody | Rabbit polyclonal anti-NDC-80 | Novus Biologicals | Novus Biologicals: 42000002; RRID:AB_10708818 | 1:200 |
| Antibody | Mouse monoclonal anti-α-tubulin | Sigma-Aldrich | Sigma-Aldrich: T6199; RRID:AB_477583 | 1:200 |
| Antibody | Mouse monoclonal anti-a-tubulin + FITC | Sigma-Aldrich | Sigma-Aldrich: F2168; RRID:AB_476967 | 1:50 |
| Antibody | Goat polyclonal anti-rabbit + Alexa Fluor 488 | Invitrogen | Invitrogen: A11034; RRID:AB_2576217 | 1:200 |
| Antibody | Goat polyclonal anti-mouse + AlexaFluor 488 | Invitrogen | Invitrogen: A11001; RRID:AB_2534069 | 1:200 |
| Antibody | Goat polyclonal anti-mouse + AlexaFluor 564 | Invitrogen | Invitrogen: A11010; RRID:AB_2534077 | 1:200 |
| Antibody | Donkey polyclonal anti-rabbit + Cy3 | Jackson ImmunoResearch | Jackson ImmunoResearch: 711-165-152; RRID:AB_2307443 | 1:500 |
| Antibody | Rabbit polyclonal anti-KNL-1 | (*Desai et al., 2003*) | | 1:500 |
| Antibody | Rabbit polyclonal anti-KNL-3 | (*Cheeseman et al., 2004*) | | 1:500 |
| Antibody | Rabbit polyclonal anti-AIR-2 | (*Schumacher et al., 1998*) | | 1:200 |
| Antibody | Rabbit polyclonal anti-CLS-2 | (*Espiritu et al., 2012*) | | 1:200 |
| Antibody | Rabbit polyclonal anti-ZEN-4 | (*Powers et al., 1998*) | | 1:200 |
| Chemical compound, drug | Polystyrene microbeads solution (0.1 µm) | Polysciences | Polysciences: 00876–15 | |

*Continued on next page*

*Continued*

| Reagent type (species) or resource | Designation | Source or reference | Identifiers | Additional information |
|---|---|---|---|---|
| Chemical compound, drug | Hexadecene | Merck | Merck: 822064 | |
| Chemical compound, drug | BSA (fraction V) | Carl Roth | Carl Roth: 8076.2 | |
| Chemical compound, drug | Osmium tetroxide | EMS | EMS: 19100 | |
| Chemical compound, drug | Uranyl acetate | Polysciences | Polysciences: 21447–25 | |
| Chemical compound, drug | Epon/Araldite epoxy resin | EMS | EMS: 13940 | |
| Chemical compound, drug | Colloidal gold (15 nm) | BBI | BBI: EM.GC15 | |
| Other | Type-A aluminum planchette | Wohlwend | Wohlwend: 241 | |
| Other | Type-B aluminum planchette | Wohlwend | Wohlwend: 242 | |
| Software, algorithm | Code for Kymograph creation | this study | | Python code provided as supplemental information |
| Software, algorithm | Code for Image volume resampling | this study | | Python code provided as supplemental information |
| Software, algorithm | arivis Vision4D | Arivis AG (https://www.arivis.com/en/imaging-science/arivis-vision4d) | | Versions 2.9–2.12 |
| Software, algorithm | IMOD | (*Kremer et al., 1996*) (https://bio3d.colorado.edu/imod/) | | Version 4.8.22 |
| Software, algorithm | ZIBAmira | (*Stalling et al., 2005*) (https://amira.zib.de/) | | Versions 2016.47–2017.55 |

## Strains and worm handling

### Strains

The following strains were used in this study: N2 wild type (*Brenner, 1974*); ANA0072 (adeIs1 [[pMD191] mex-5p::spd-1::GFP + unc-119(+)] II; unc-119(ed3) III; ltIs37 [(pAA64) pie-1p::mCherry::his-58 + unc-119(+)] IV) (*Nahaboo et al., 2015*); CB1489 (him-8(e1489) IV) (*Herman and Kari, 1989*; *Phillips et al., 2005*); CB2580 (tra-2(e1094)/dpy-10(e128) II) (*Hodgkin, 1985*); MAS91 (unc-119(ed3) III; ltIs37[pAA64; pie-1::mCherry::HIS58]; ruIs57[pie-1::GFP::tubulin + unc-119(+)]) (*Han et al., 2015*); MAS96 (unc-119(ed3) III; ddIs6[tbg-1::GFP + unc-119(+)]; ltIs37[pAA64; pie-1::mCherry::HIS-58 + unc-119(+)] IV, qaIs3507[pie-1::GFP::LEM-2 + unc-119(+)]) (M. Srayko, Alberta); TMR17 (unc-119 (ed3) III; ddIs6[tbg-1::GFP + unc-119(+)]; ltIs37[pAA64; pie-1::mCherry::HIS-58 + unc-119(+)] IV) (this study); TMR18 (him-8(e1489) IV; unc-119(ed3) III; ddIs6[tbg-1::GFP + unc-119(+)]; ltIs37[pAA64; pie-1::mCherry::HIS-58 + unc-119(+)] IV) (this study); TMR26 (zim-2 (tm574) IV; unc-119(ed3) III; ddIs6 [tbg-1::GFP + unc-119(+)]; ltIs37[pAA64; pie-1::mCherry::HIS-58 + unc-119(+)] IV) (this study); XC110 (tra-2(e1094)/dpy-10(e128) II; unc-119(ed3) III; ltIs37[pAA64; pie-1::mCherry::HIS58] (IV); ruIs57[pie-1::GFP::tubulin + unc-119(+)]) (this study); XC116 (tra-2(e1094)/dpy-10(e128) II; ddIs6[tbg-1::GFP + unc-119(+)]; ltIs37[pAA64; pie-1::mCherry::HIS-58 + unc-119(+)] IV) (this study); SP346 (tetraploid, 4 n) (*Madl and Herman, 1979*).

### Worm handling

Worms were grown on nematode growth medium (NGM) plates at 20°C with *E. coli* (OP50) as food source (*Brenner, 1974*). Male worms were produced by exposing L4 hermaphrodites to 30°C for 4–6 h and checking the resulting progeny for male worms after three days (*Sulston and Hodgkin, 1988*).

Males were maintained by mating 20–30 male worms with five L4 hermaphrodites. Triploid worms were obtained by mating tetraploid hermaphrodites with males of either MAS91 or TMR17. F1 male animals were selected and imaged as described below.

## Light microscopy and analysis of spindle dynamics

### Light microscopy

Age-synchronized males (3 days after bleaching adult hermaphrodites fertilized by males) were placed in droplets of 1 µl polystyrene microbeads solution (diameter of 0.1 µm; Polysciences, USA) on 10% agarose pads. Samples were then covered with a coverslip and sealed with wax (*Kim et al., 2013*). We used a confocal spinning disk microscope (IX 83, Olympus, Japan) equipped with a 60 × 1.2 NA water immersion objective and an EMCCD camera (iXon Ultra 897, Andor, UK) for live-cell imaging. The meiotic region within single males was imaged for about one hour and a z-stack was recorded either every 20 s or 30 s. Images were then corrected for photobleaching using the Fiji software package (*Schindelin et al., 2012*).

### Analysis of spindle dynamics

Image stacks were analyzed with the arivis Vision4D software package (arivis AG, Germany). Individual spindles were cropped and spindle poles in each frame were segmented by thresholding. The Euclidean distance of the center of mass of both spindle poles was then calculated for each time point. To produce kymographs, the original image data were resampled with a custom-made python script in arivis Vision4D (*Source code 1*). The spindle axes were rotated in all three dimensions to align the axis along the z-direction. As a consequence, each spindle had a comparable orientation with an isotropic voxel size of 0.1 µm and a radius of 0.9 µm around the spindle axis. All voxels were then recalculated based on the initial transformation of the axis with an extrapolation of 1 µm at each pole in the direction of the axis. As the axes of the spindles were chosen to lay in the z-dimension all images in the resampled datasets were laying orthogonally to it (x, y-plane). For the calculation of kymographs, the Gaussian weighted sum of fluorescence was calculated in each plane in 0.1 µm steps along this axis and repeated for all time points. For the analysis of chromosome movements, the two peak maxima from the kymographs of the chromosome and spindle pole fluorescence signals were then used to calculate the respective distances for each time point. These distances were then plotted against time relative to the onset of anaphase and utilized to determine the spindle characteristics, that is the pole-to-pole and autosome-to-autosome distance, the speed of segregation and the time of spindle elongation (*Table 2*).

Individual measurements were aligned according to the onset of anaphase and the mean distance was then calculated and plotted against time relative to anaphase onset. For characterizing the dynamic properties of spindles these mean values were then used to determine spindle length at metaphase and after anaphase. The initial speed of spindle elongation and chromosome movement was calculated by fitting a linear function to the measurements during the first minute after anaphase onset as the segregation speed slowed down continuously afterwards.

To illustrate the process of division, the spindles were resampled and rotated as described above but with a radius of 3 µm around the spindle axis and an extrapolation of 2 µm after the spindle poles (*Source code 2*). Then a y,z-projection over x (maximum intensity) was calculated for each time point to display the resampled volume as a plane image (*Figures 1–2* and *6–7*). For a comparison of microtubule density on both sides of the X chromosome facing the spindle poles, the sum of fluorescence was calculated within two cubic boxes (with a similar volume of 1 µm$^3$) adjacent to the X chromosome in the resampled light microscopic image data. The box on the side, where the chromosome moved to at the time of segregation, was termed 'volume 1', the other 'volume 2'. The ratio between both values at each time point indirectly describes the difference in the number of microtubules (*Figure 6E–F*).

For each data set, the visco-elastic property of the X chromosome was probed by segmenting it in a resampled 3D dataset and measuring its dimensions. Along the spindle axis, the length of the X chromosome was measured (z-dimension). Orthogonal to the z-axis, the mean values for the x- and y-dimension were calculated. A shape coefficient was then calculated (z/[(x+y)/2]) to illustrate the change of the shape of the X chromosome over time (*Figure 2A–B*).

The centrosomes were segmented in 3D image data from worms expressing γ-tubulin::GFP and histone H2B::mCherry with the arivis Vision4D software package by applying a cut-off threshold to the 3D image data. All fluorescence signals above the threshold were included in the segment of the centrosomes. The volume of the segments was then calculated for each frame and each centrosome individually for spindles in meiosis I and II. When centrosomes split in meiosis I and could be segmented individually both volumes were summed together for the respective frame (*Appendix 1—figure 3*).

## Immunostaining for light microscopy

For antibody staining of *C. elegans* gonads, synchronized males were dissected and fixed in 1% paraformaldehyde using established protocols (*Howe et al., 2001*). Methanol/acetone fixation was used for immunolabeling of mitotic and meiotic embryos (*Shakes et al., 2009*). Primary and secondary antibodies were diluted in blocking buffer (PBS + 0.1% Tween 20 and 10 mg/ml BSA) and staining was conducted at room temperature in a humid chamber. Primary antibodies were used in overnight incubations (unless otherwise noted). Commercial sources or labs kindly providing antibodies were as listed: 1:200 rabbit anti-NDC-80 (Novus Biologicals, catalog #42000002); 1:200 mouse anti-α-tubulin (DM1A Sigma-Aldrich, catalog #T6199); 1:500 rabbit anti-KNL-1 (*Desai et al., 2003*); 1:500 rabbit anti-KNL-3 (*Cheeseman et al., 2004*); 1:200 rabbit anti-AIR-2 (*Schumacher et al., 1998*); 1:200 rabbit anti-CLS-2 (*Espiritu et al., 2012*); 1:200 rabbit anti-ZEN-4 (*Powers et al., 1998*); and 1:50 FITC-conjugated anti-α-tubulin (Sigma-Aldrich, #F2168). Secondary antibodies included: goat anti-rabbit AlexaFluor 488-labeled IgG (used at 1:200); goat anti-mouse AlexaFluor 488-labeled IgG (used at 1:200); goat anti-mouse AlexaFluor 564-labeled IgG (used at 1:200); and donkey anti-rabbit Cy3 (used at 1:500). DNA was visualized using DAPI at 0.1 µg/ml. Slides were prepared by using VectaShield (Vector Labs, USA) as a combined mounting and antifade medium. Confocal images were acquired using a Zeiss LSM710 microscope, a Zeiss LSM880 microscope, or a Leica SP8 Confocal System (*Figure 2E* and *Appendix 1—figure 8A*). Super-resolution images were collected using an OMX 3D-SIM microscope (GE Healthcare, USA) with an Olympus (Shinjuku, Japan) 100x UPlanSApo 1.4 NA objective (Olympus, Japan). Images were captured in z-steps of 0.125 µm and processed using SoftWoRx (GE Healthcare, USA) and IMARIS (Bitplane, Switzerland) 3D imaging software (*Appendix 1—figure 1*).

## Laser microsurgery

Age-synchronized males (3 days old) were placed within a of droplet of 1 µl M9 buffer containing 1 mM levamisole and 0.1 µm polystyrene microbeads (Polysciences, USA) on a 10% agarose pad. Samples were then covered with a coverslip and sealed with wax. For imaging during laser microsurgery, we used a confocal spinning disk microscope (Ti Eclipse, Nikon, Japan) equipped with a 60 × 1.2 NA water immersion objective, a 1.5x optovar, an EMCCD camera (iXon Ultra 897, Andor, UK) and a mode-locked femtosecond Ti:sapphire laser (Chameleon Vision II, Coherent, USA) operated at a wavelength of 800 nm. After locating spindles in anaphase I within males, a single image was recorded in intervals of 1 s. Subsequently, a position for the laser cut was chosen and a single spot with a diameter of about 1.3 µm was ablated with a laser power of 150 mW and an exposure time of 30 ms. Image acquisition was continued until the X chromosome had been fully segregated (*Figure 2C–D*). In total, we performed 26 laser ablations. In ~80% of the experiments we observed a movement of the X towards the unablated side. For further analysis the images were corrected for photobleaching within the Fiji software package and corrected for movement using the plugin 'image stabilizer' (http://www.cs.cmu.edu/~kangli/code/Image_Stabilizer.html; February 2008).

## Specimen preparation for electron microscopy

Males were ultra-rapidly frozen using an HPF COMPACT 01 high-pressure freezer (Engineering Office M. Wohlwend, Sennwald, Switzerland). For each freezing run, five individuals were placed in a type-A aluminum planchette (100 µm deep; Wohlwend, article #241) pre-wetted with hexadecene (Merck) and then filled with M9 buffer containing 20% (w/v) BSA (Roth, Germany). The specimen holders were closed by gently placing a type-B aluminum planchette (Wohlwend, article #242) with the flat side facing the sample on top of a type-A specimen holder. The sandwiches were frozen under high pressure (~2000 bar) with a cooling rate of ~20000 °C/s (*Fabig et al., 2019*). Specimen

holders were opened under liquid nitrogen and transferred to cryo-vials filled with anhydrous ace-
tone containing 1% (w/v) osmium tetroxide (EMS) and 0.1% (w/v) uranyl acetate (Polysciences, USA).
Freeze substitution was performed in a Leica AFS (Leica Microsystems, Austria). Samples were kept
at −90℃, then warmed up to −30℃ with steps of 5 ℃/h, kept for 5 h at −30℃ and warmed up again
(steps of 5 ℃/h) to 0℃. Subsequently, samples were washed three times with pure anhydrous ace-
tone and infiltrated with Epon/Araldite (EMS, USA) epoxy resin at increasing concentrations of resin
(resin:acetone: 1:3, 1:1, 3:1, then pure resin) for 2 h each step at room temperature (*Müller-
Reichert et al., 2003*). Samples were incubated with pure resin over night and then for 4 hr. Samples
were thin-layer embedded between two Teflon-coated glass slides and allowed to polymerize at 60°
C for 48 h (*Müller-Reichert et al., 2008*). Polymerized samples were remounted on dummy blocks
and semi-thin serial sections (300 nm) were cut using an EM UC6 (Leica Microsystems, Austria) ultra-
microtome. Ribbons of sections were collected on Formvar-coated copper slot grids, post-stained
with 2% (w/v) uranyl acetate in 70% (v/v) methanol and 0.4% (w/v) lead citrate and allowed to dry
prior to inspection.

## Electron tomography, microtubule segmentation and stitching of data sets

In preparation for electron tomography, both sides of the samples were coated with 15 nm-colloidal
gold (BBI, UK). To select cells in meiosis, serial sections were pre-inspected at low magnification
(~2900 x) using a Zeiss EM906 transmission electron microscope (Zeiss, Germany) operated at 80 kV.
Serial sections containing cells/regions of interest were then transferred to a Tecnai F30 transmission
electron microscope (Thermo Fischer Scientific, USA) operated at 300 kV and equipped with a
US1000 CCD camera (Gatan, USA). Tilt series were acquired from −65° to +65° with 1° increments at
a magnification of 4700x (pixel size 2.32 nm). Specimens were then rotated 90° to acquire a second
tilt series for double-tilt electron tomography (*Mastronarde, 1997*). Electron tomograms were calcu-
lated using the IMOD software package (*Kremer et al., 1996*). As previously described
(*Redemann et al., 2014*; *Weber et al., 2012*), microtubules were automatically segmented using
the ZIBAmira (Zuse Institute Berlin, Germany) software package (*Stalling et al., 2005*).

Individual tomograms were then stitched and combined (*Weber et al., 2014*) to represent whole
microtubule networks in 3D models (*Redemann et al., 2017*). Chromosomes, kinetochores and cen-
trioles were manually segmented. Kinetochores were modeled around each chromosome by gradu-
ally increasing the chromosome volume until the area of the ribosome-free zone around each
chromosome (*Howe et al., 2001*; *O'Toole et al., 2003*) was covered, giving a thickness of the male
meiotic holocentric kinetochore of about 150 nm (*Figures 3–4*, *Appendix 1—figures 4* and *7*).

## Analysis of tomographic data

### Staging of tomographic data sets

For staging of the reconstructed spindles, we determined the autosome-to-autosome distance. We
measured the distance of the individual chromosome pairs, calculated the mean of these individual
distances and ordered them accordingly. As an additional criterion for staging, we took the ‚state'
of the centrosome into account, as the centrioles pre-early split in *C. elegans* male meiosis (see
*Table 1*). Within each data set, the distance between the mother and the daughter centriole was
determined at each spindle pole and averaged. As an example, this read-out was used to determine
anaphase onset.

### Classification of microtubules

First, the distance between each point of a microtubule segment and the closest point of the surface
of individual chromosomes was calculated. Only microtubules within a distance of 150 nm or less
were considered kinetochore microtubules as this distance was measured to be the approximate
extent of the kinetochore in the electron tomograms. The kinetochore is visible in the electron tomo-
grams as a less stained region around the chromosomes (*Howe et al., 2001*). Additionally, each
kinetochore microtubule was assigned to the X chromosome or to one of the autosomal chromo-
somes according to its closest distance to the chromosome surface. As microtubules in anaphase
pass between the autosomes and attach to the X chromosome after that, they were first checked for
an association with the X chromosome and if there was none, further analysis was performed to

check for a potential autosomal association. For each chromosome the microtubule associations were subdivided between end-on and lateral (*Video 1*). We defined an end-on association by extrapolating the microtubule after its end for 150 nm and checking if this extrapolated line was cutting the surface of the chromosome. If that criterion was not met, we considered the association of the microtubule with the given chromosome as lateral.

### Length distribution

Furthermore, we analyzed the length distribution of microtubules. For each microtubule class in each meiotic spindle the length distribution is given (mean, standard deviation). Further, the variance among the datasets was compared using a one-way analysis of variance (ANOVA; *Appendix 1—figure 5*).

We also analyzed the ratio of the sum of microtubule length between two defined volumes analogous to the analysis of the light microscopic data. For that a box of 1 $\mu m^3$ was placed on either side of the X chromosome facing the spindle poles. The microtubules within this box were extracted and their length was measured and summed up. The ratio of the box closer to the respective pole against the second box was calculated (*Figure 6A–B*). The microtubule tortuosity (microtubule spline length divided by end-end length; *Appendix 1—figure 6*) was measure for end-on and lateral microtubules in contact with the X chromosome.

### Chromosome shape

Further, we analyzed the shape of the chromosomes in the EM data as previously described (*Lindow et al., 2018*). In brief, chromosomes were manually segmented and along the pole-to-pole axis of the spindle orthogonal planes were placed with 10 nm spacing. For every plane the area was calculated that intersected the individual chromosome surface. After plotting the cross-sectional area against the pole-to-pole distance a Gaussian function containing five terms was fit with MATLAB (MATLAB 2017b, The MathWorks, USA) and the full width at half maximum (FWHM) of the for each chromosome was determined and compared (*Figure 8A–B*). For measuring the distance between centrioles and the end-on microtubule end at the autosomes, we first selected the closest centriole at the putative microtubule minus-end. Then we extracted the position of the respective putative plus-end and calculated the Euclidean distance between the centriole and the putative plus-end (*Figure 8C–D*). The angle between the microtubule plus-end and the chromosome-centrosome axis was determined by calculating the vector between the respective chromosome and the centrosome and the vector between chromosome and the respective microtubule plus-end. Then the angle between both vectors was calculated (*Figure 8E–F*).

## Acknowledgements

The authors would like to thank Dr. Michael Laue (Robert Koch Institute, Berlin, Germany) for using the COMPACT 01 (Wohlwend) high-pressure freezer, the Core Facility Cellular Imaging of the Faculty of Medicine Carl Gustav Carus (TU Dresden, Germany) and the light- and electron microscopy facilities at the MPI-CBG (Dresden, Germany) for technical assistance. The Delattre, Desai, Rose, Schumacher, and Strome labs generously provided strains or antibodies used for these studies. We are also grateful to Drs. Diane Shakes (Williamsburg VA, USA), Stefanie Redemann (Charlottesville VA, USA) and Kevin O'Connell (Bethesda MD, USA) for a critical reading of the manuscript. We would like to thank Martin Merkel, Ewa Kania, Sophia Merkel, Maura Hofmann and Isabelle Kunert for help in tomographic reconstruction and microtubule segmentation. The authors are grateful to Falko Löffler, Carola Bender and Christian Götze (arivis AG) for help with image processing in *arivis Vision4D.* Some strains were provided by the CGC, which is funded by NIH Office of Research Infrastructure Programs (P40 OD010440). We acknowledge NIH grant NIH1S10OD024988-01 for the purchase on the OMX microscope. Research in the Müller-Reichert laboratory is supported by the Deutsche Forschungsgemeinschaft (MU 1423/10–1). RK received funding from the European Union's Horizon 2020 research and innovation program under the Marie Skłodowska-Curie grant agreement No. 675737 (grant to TMü-R). Work in the Chu lab is supported by the NIH grant R03 HD093990-01A1 and the NSF Awards RUI-1817611 and DBI-1548297.

## Additional information

### Funding

| Funder | Grant reference number | Author |
|---|---|---|
| Deutsche Forschungsge-meinschaft | MU 1423/10-1 | Gunar Fabig<br>Thomas Müller-Reichert |
| H2020 Marie Skłodowska-Curie Actions | No. 675737 | Robert Kiewisz<br>Thomas Müller-Reichert |
| National Institutes of Health | R03 HD093990-01A1 | Vanessa Cota<br>Diana S Chu |
| National Science Foundation | RUI-1817611 | Vanessa Cota<br>Diana S Chu |
| National Institutes of Health | NIH1S10OD024988-01 | James A Powers |
| National Science Foundation | DBI-1548297 | Vanessa Cota<br>Diana S Chu |

The funders had no role in study design, data collection and interpretation, or the decision to submit the work for publication.

### Author contributions

Gunar Fabig, Conceptualization, Data curation, Formal analysis, Investigation, Visualization, Methodology, Writing - original draft, Writing - review and editing, Live imaging, electron tomography, reconstructions, data analysis, creation of Figures; Robert Kiewisz, Formal analysis, Reconstruction and analysis of electron tomograms; Norbert Lindow, Software, Methodology, Software and analysis tool development for electron tomograms in ZIB Amira; James A Powers, Resources, Formal analysis, Acquisition of super-resolution data with the DeltaVision OMX microscope (Appendix-Figure 1); Vanessa Cota, Formal analysis, Immunostainings (Fig. 2E); Luis J Quintanilla, Formal analysis, Immunostainings (Appendix-Figure 8A); Jan Brugués, Resources, Methodology, Advice with the laser ablation experiments (Fig. 2C and D); Steffen Prohaska, Software, Funding acquisition, Software and analysis tool development for electron tomograms in ZIB Amira; Diana S Chu, Conceptualization, Supervision, Funding acquisition, Methodology, Writing - review and editing, Co-supervision; Thomas Müller-Reichert, Conceptualization, Supervision, Funding acquisition, Methodology, Writing - original draft, Co-supervision

### Author ORCIDs

Gunar Fabig (iD) https://orcid.org/0000-0003-3017-0978
Robert Kiewisz (iD) http://orcid.org/0000-0003-2733-4978
Diana S Chu (iD) https://orcid.org/0000-0002-4653-7909
Thomas Müller-Reichert (iD) https://orcid.org/0000-0003-0203-1436

### Decision letter and Author response

Decision letter https://doi.org/10.7554/eLife.50988.sa1
Author response https://doi.org/10.7554/eLife.50988.sa2

## Additional files

### Supplementary files

• Source code 1. Kymograph. This script needs two 3D coordinates and 3D image data as an input and creates a 2D line scan in between the two input coordinates according to the settings in the script. If a time series is used this algorithm creates a kymograph.

• Source code 2. Volume resampling. This script needs two 3D coordinates and 3D image data as an input and creates a spatially reoriented and resampled 3D data set in between the two input coordinates according to the settings in the script. The center line (spindle axis) in between the two input

coordinates is positioned in the z-dimension of the output data set. If a time series is used this algorithm creates a resampled 3D data set over time.

- Transparent reporting form

## Data availability

Data have been uploaded to the TU Dresden Open Access Repository and Archive system (OpARA) and are available as open access: https://doi.org/10.25532/OPARA-56.

The following dataset was generated:

| Author(s) | Year | Dataset title | Dataset URL | Database and Identifier |
|-----------|------|---------------|-------------|-------------------------|
| Fabig G | 2020 | Supplemental data for the publication | http://dx.doi.org/10.25532/OPARA-56 | OpARA, 10.25532/OPARA-56 |

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

## Appendix 1

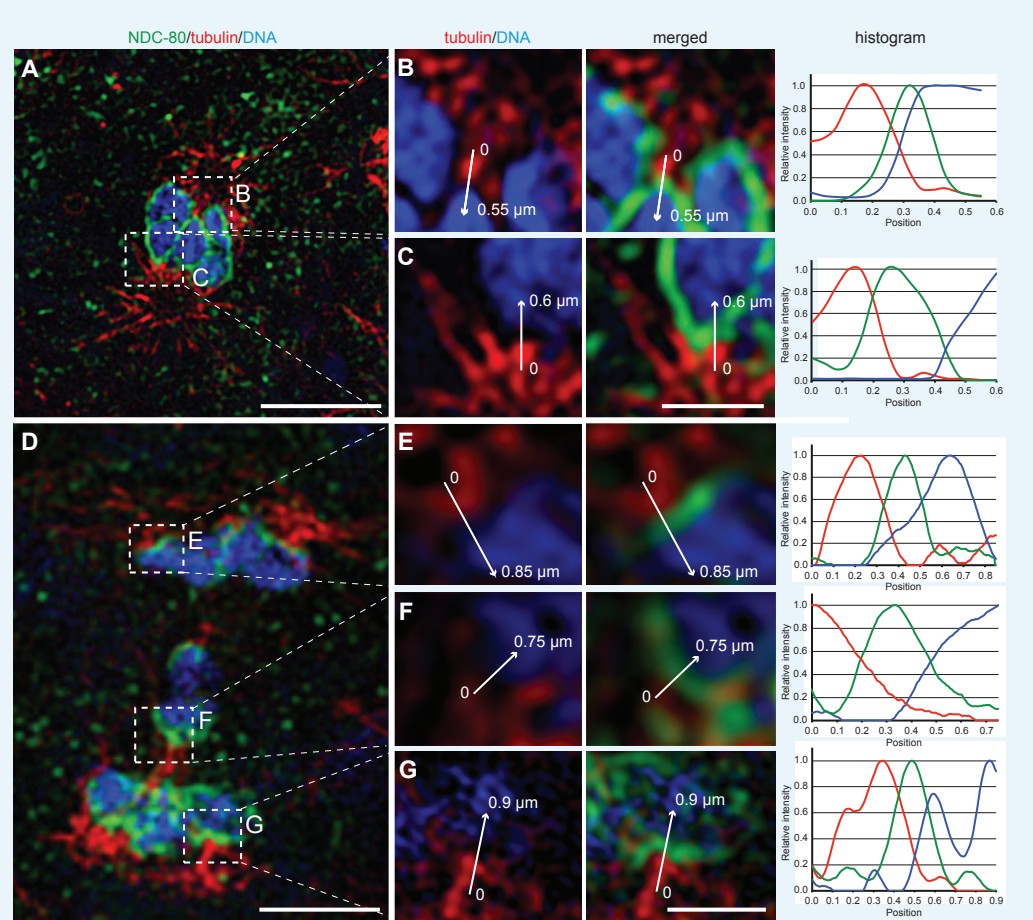

**Appendix 1—figure 1.** The outer kinetochore protein NDC80 localizes between chromosomes and microtubules at spermatocyte metaphase and anaphase I. (**A**) Super-resolution fluorescence microscopy of metaphase I in fixed *him-8*(e1489) X0 males stained with antibodies against α-tubulin (red) and NDC-80 (green). DAPI stained DNA is in blue. Scale bar, 2 μm. (**B–C**) Enlargement of boxed regions as shown in (A) highlighting microtubule and NDC-80 localization relative to metaphase chromosomes. Normalized intensity values along the arrows for each staining pattern are plotted in the histograms (right panels). Scale bars, 0.5 μm. (**D**) Super-resolution fluorescence microscopy of anaphase I in *him-8* X0 males. Imaging conditions were as given in (A). Scale bar, 2 μm. (**E–G**) Enlargement of boxed regions as shown in (A) highlighting microtubule and NDC-80 localization relative to separating chromosomes. Scale bar, 0.5 μm.

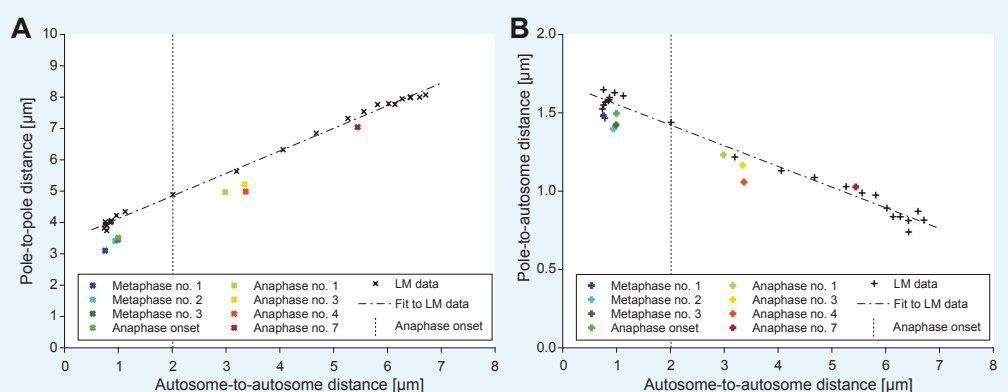

**Appendix 1—figure 2.** Comparison of electron tomographic and light microscopic data. (**A**) Pole-to-pole distance plotted against the autosome-to-autosome distance for each tomographic data set (color coding is as shown in *Figure 6B*). Light microscopic measurements (*Figure 7E*) are shown in black. For the purpose of staging, the plot illustrates where the tomographic data sets are 'positioned' with respect to the averaged data from light microscopy (fitted dashed line in black) obtained. (**B**) Pole-to-autosome distance plotted against the autosome-to-autosome distance.

The online version of this article includes the following source data is available for figure 2:

**Appendix 1—figure 2—source data 1.** Measurements of autosome-to-autosome and pole-to-autosome distances from replicates used in *Appendix 1—figure 2*.

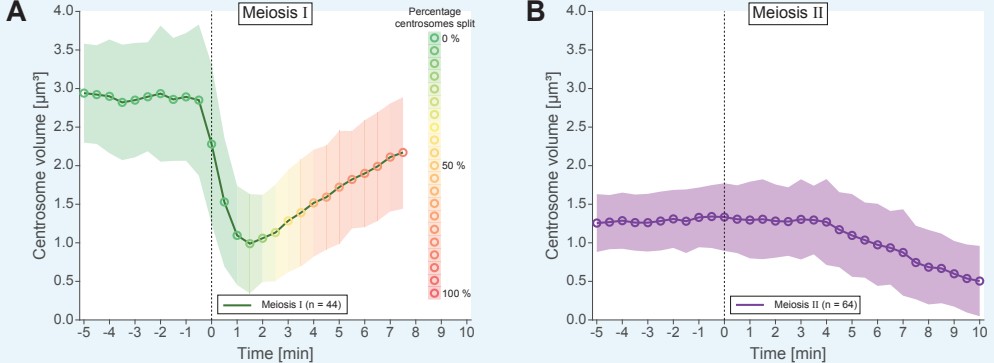

**Appendix 1—figure 3.** Analysis of centrosomal volumes in meiosis I and II. (**A**) Plot showing centrosome volume in meiosis I over time. The 3D volume was measured in worms expressing γ-tubulin::GFP and histone H2B::mCherry. For each dataset a fixed threshold was defined to segment the outer border of the centrosome. The mean volume is plotted as a green line for unsplitted centrosomes and shown as an orange to red line for splitted centrosomes. The percentage of splitted centrosomes is indicated by this color change. For splitted centrosomes, the sum of both separated centrosomes was determined (n = 44). The standard deviation is depicted as a shaded area. (**B**) Centrosome volume over time (purple line) in meiosis II (n = 64).

The online version of this article includes the following source data is available for figure 3:

**Appendix 1—figure 3—source data 1.** Measurements of centrosome volume from replicates used in *Appendix 1—figure 3*.

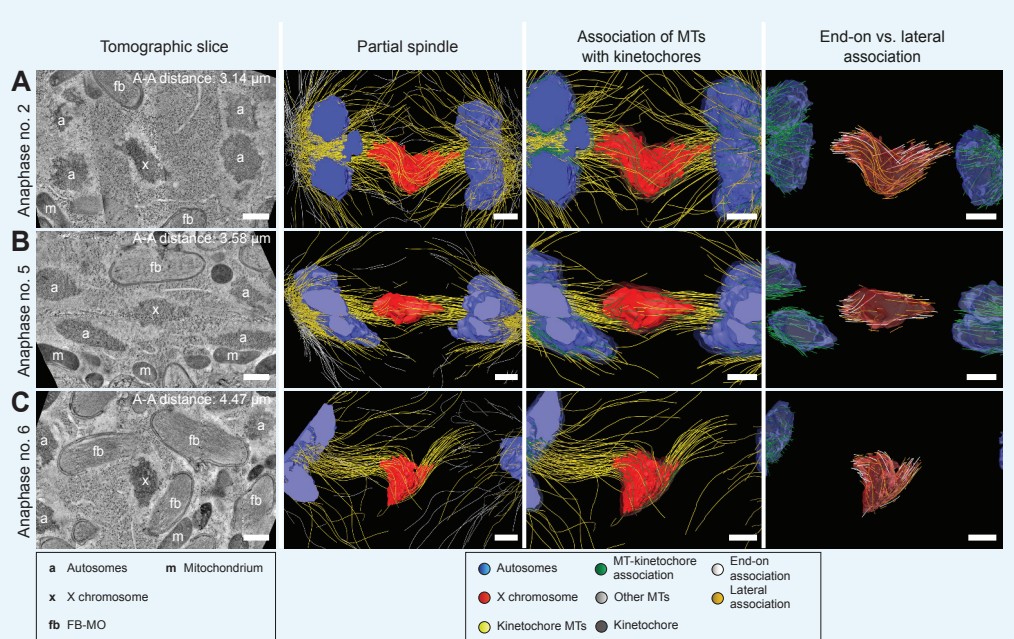

**Appendix 1—figure 4.** Visualization of partially reconstructed spindles in mid/late anaphase I. (**A**) Mid anaphase spindle with a pole-to-pole distance of 3.14 µm. (**B**) Mid anaphase spindle with a pole-to-pole distance of 3.58 µm. (**C**) Mid anaphase spindle with a pole-to-pole distance of 4.47 µm. Left panels: tomographic slice showing the autosomes (a), and the univalent X chromosome (x) aligned along the spindle axis. Mitochondria (m) and fibrous body-membranous organelles (fb) are also indicated. Mid left panels: corresponding three-dimensional model illustrating the organization of the partially reconstructed spindle. Autosomes are in blue, the X chromosome in red, microtubules within a distance of 150 nm or closer to the chromosome surfaces in yellow and all other microtubules in gray. Mid right panels: association of microtubules with the kinetochores. Kinetochores are shown as semi-transparent regions around each chromosome. The part of each microtubule entering the kinetochore region around the holocentric chromosomes is in green. Right panels: visualization of end-on (white) *versus* laterally (orange) X chromosome-associated microtubules. The part of each microtubule entering the kinetochore region around the holocentric chromosomes is shown in green. The autosome-to-autosome distance (A-A) for each reconstruction is indicated in the left column. Scale bars, 500 nm.

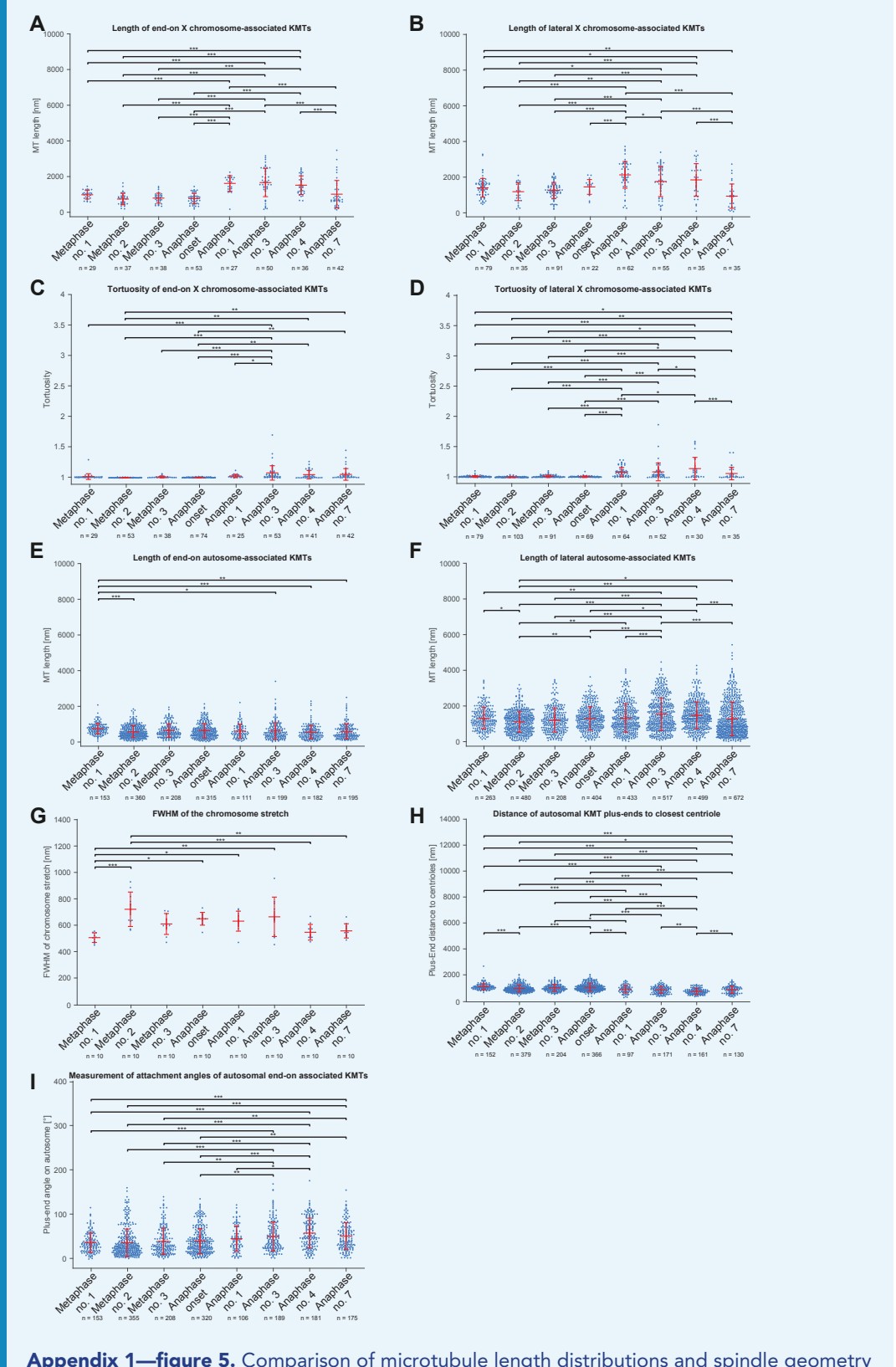

**Appendix 1—figure 5.** Comparison of microtubule length distributions and spindle geometry (ANOVA). (**A**) Length of end-on X chromosome-associated kinetochore microtubule for each tomographic data set as shown in *Figure 5A*. The mean, the standard deviation and single measurements for each meiotic stage are given. Results of a one-way analysis of variance

(ANOVA) of all data sets against each other are shown. Level of significance: * is p<=0.05; ** is p<=0.01; and *** is p<=0.001. (B) Length of lateral X chromosome-associated kinetochore microtubules corresponding to *Figure 5B*. (C) Tortuosity of end-on X chromosome-associated kinetochore microtubules corresponding to *Appendix 1—figure 6B*. (D) Tortuosity of lateral X chromosome-associated kinetochore microtubules corresponding to *Appendix 1—figure 6B*. (E) Length of end-on autosome-associated kinetochore microtubules corresponding to *Figure 7G*. (F) Length of lateral autosome-associated kinetochore microtubules corresponding to *Figure 7H*. (G) FWHM of the chromosome stretch for each meiotic stage corresponding to *Figure 8B*. (H) Distance of autosomal kinetochore microtubule plus-ends to closest centriole for each meiotic stage corresponding to *Figure 8D*. (I) Measurement of attachment angles of autosomal end-on associated kinetochore microtubules as given in *Figure 8F*.

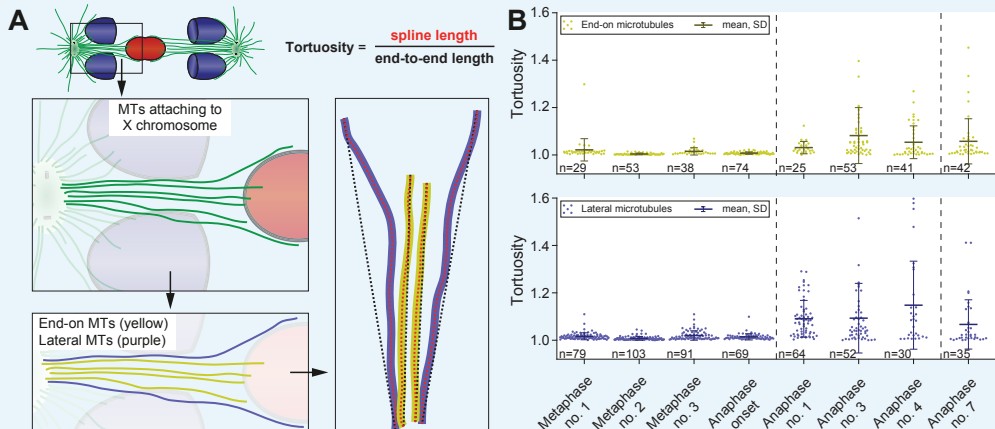

**Appendix 1—figure 6.** Tortuosity of X chromosome-associated microtubules during anaphase I. (A) Schematic drawing illustrating the shape of X chromosome-associated kinetochore microtubules. Both end-on (yellow) and laterally associated microtubules (purple) are shown (left panels). The tortuosity of each microtubule is given by the spline length (red dotted lines; right panel) divided by the end-to-end length (black dotted lines). Dashed lines indicate a grouping of the spindles according to the meiotic stages: metaphase/anaphase onset, mid anaphase and late anaphase (see also *Appendix 1—figure 2*). (B) Plots showing the tortuosity of end-on (top) and laterally (bottom) associated kinetochore microtubules. The meiotic stages correspond to the data sets as shown in *Figures 3* and *4*. Mean, standard deviation and individual measurements are given for each data set.

The online version of this article includes the following source data is available for figure 6:

**Appendix 1—figure 6—source data 1.** Measurements of tortuosity of end-on and lateral microtubules from replicates used in *Appendix 1—figure 6*.

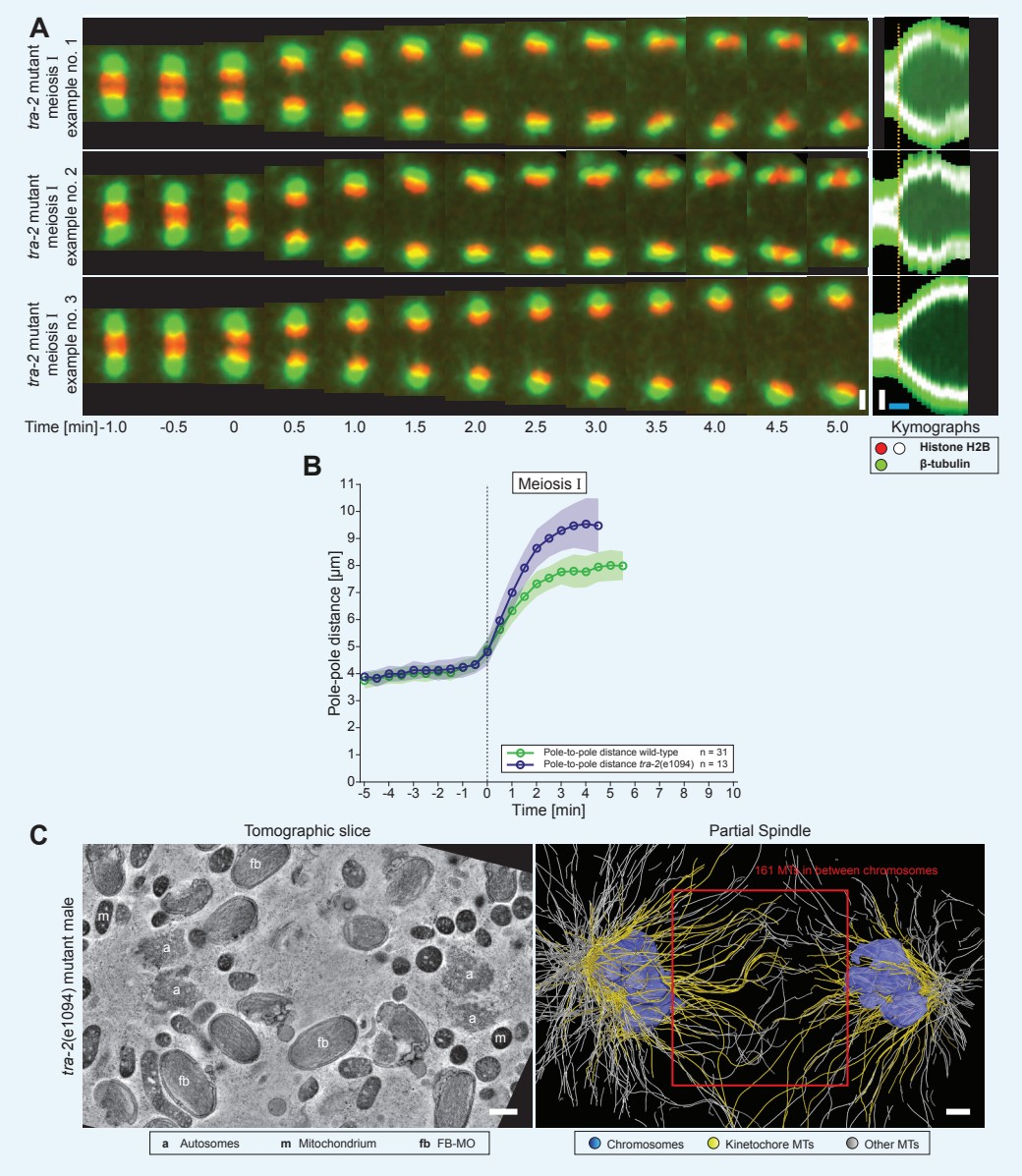

**Appendix 1—figure 7.** Organization and dynamics of *tra-2*(e1094) spindles in meiosis I. (**A**) Time series of confocal image projections of three examples of *tra-2*(e1094) mutant spindles in meiosis I. Microtubules (β-tubulin::GFP) and chromosomes (histone H2B::mCherry) are visualized in green and red, respectively. Anaphase onset is time point zero (t = 0). Chromosome segregation is also visualized in kymographs (right panels). The start of the separation of the autosomes is indicated by a dashed line (orange). Scale bars (white), 2 μm; time bar (blue), 2 min. (**B**) Quantitative analysis of spindle elongation (pole-to-pole distance) over time in meiosis I in wild-type (green) and *tra-2*(e1094) (blue) spindles. Anaphase onset is time point zero (t = 0). The mean and standard deviation is given (circles and shaded areas). (**C**) Tomographic slice through a partial *tra-2*(e1094) mutant spindle in anaphase I (left panel). Autosomes (a), mitochondria (m) and FB-MOs (fb) are indicated. Three-dimensional model of the same *tra-2*(e1094) mutant spindle (right panel). Chromosomes are in blue, microtubules within a distance of 150 nm or closer to the chromosome surfaces in yellow and all other microtubules in white. The spindle midzone is indicated (red rectangle). Scale bars, 500 nm.

The online version of this article includes the following video and source data for figure 7:

**Appendix 1—figure 7—source data 1.** Measurements of pole-to-pole distance over time in replicates used in *Appendix 1—figure 7B*.

**Appendix 1—figure 7—video 1.** Organization of *tra-2* spindles in meiosis I. This video shows a three-dimensional model of a *tra-2*(e1094) mutant spindle at anaphase I. Chromosomes are shows in blue, kinetochore microtubules in yellow, other microtubules in white. This video corresponds to *Appendix 1—figure 7C*.

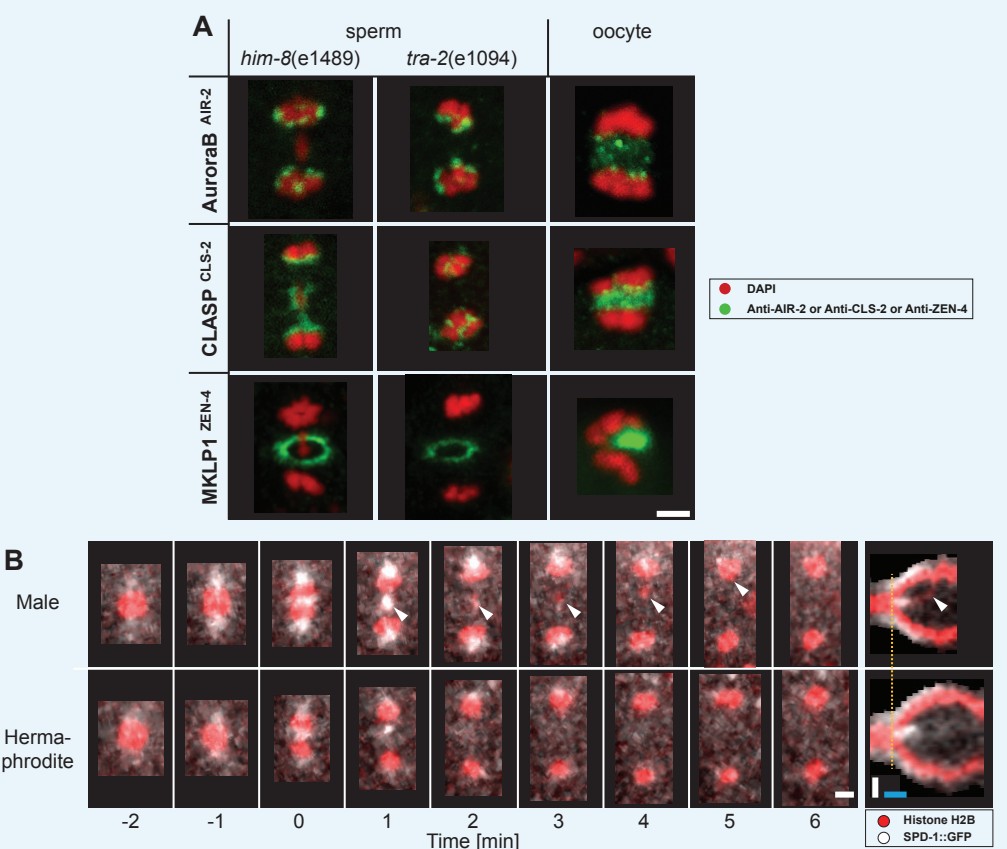

**Appendix 1—figure 8.** Composition of the spindle midzone in male meiosis I. (**A**) Confocal light microscopy of the spindle midzone in *him-8*(e1489) and *tra-2*(e1094) spermatocytes. For comparison, the spindle midzone in wild-type oocytes in anaphase I is also shown. Samples were stained with antibodies against three different proteins: AuroraB[AIR-2], CLASP[CLS-2] and MKLP1[ZEN-4]. Stained proteins are shown in green, chromosomes in red. Scale bar, 2 μm. (**B**) Time series of confocal image projections of wild-type meiosis I in males (upper row) and hermaphrodite spermatocytes (lower row). The microtubule bundling protein PRC1[SPD-1] (SPD-1::GFP, white) and the chromosomes (histone H2B::mCherry, red) are visualized. Anaphase onset is time point zero (t = 0). White arrowheads mark the position of the unpaired X chromosome in meiosis I. Chromosome segregation is also visualized in kymographs (right panels; the start of the separation of the autosomes is indicated by a dashed orange line). Scale bars (white), 2 μm; time bar (blue), 2 min.

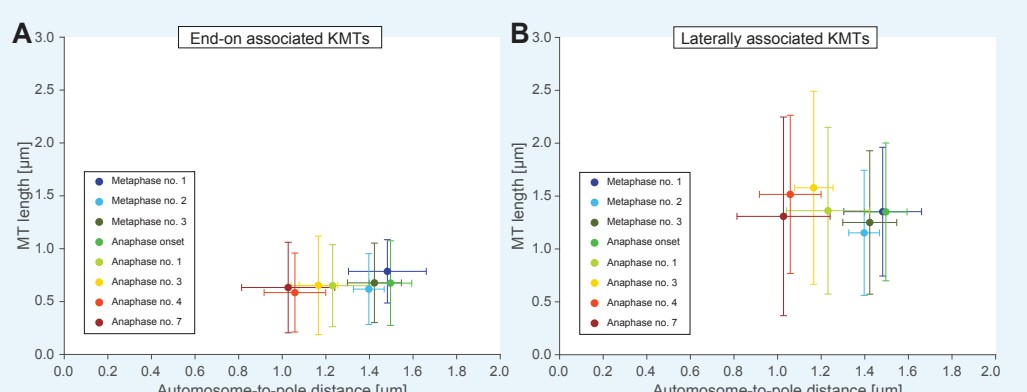

**Appendix 1—figure 9.** Length of end-on and laterally autosome-associated microtubules. (**A**) Mean length of end-on associated microtubules plotted against the autosome-to-pole distance for all fully reconstructed spindles (as given in *Figures 3* and *4*). The plot shows the mean values with standard deviations. (**B**) Identical analysis for laterally associated microtubules.

The online version of this article includes the following source data is available for figure 9:

**Appendix 1—figure 9—source data 1.** Measurements of pole-to-autosome and microtubule length in each stage shown in *Figures 3* and *4* used in Appendix1-figure 9-source data1.

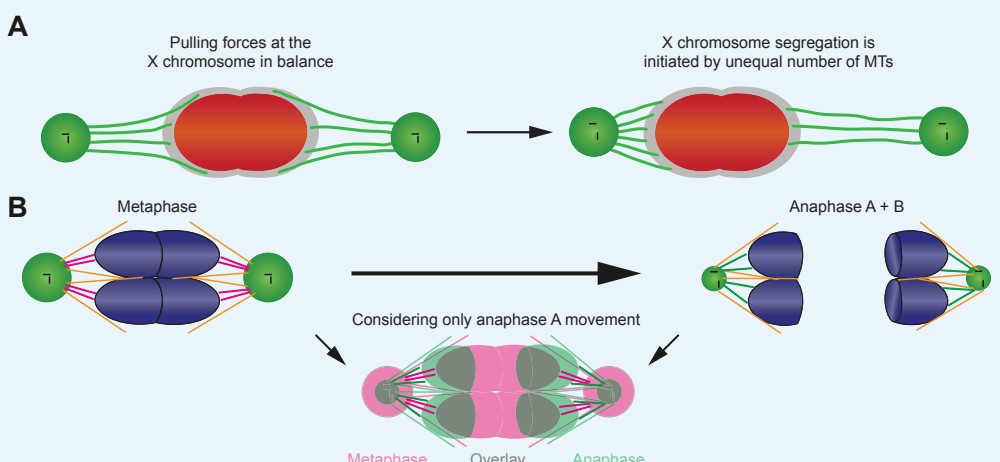

**Appendix 1—figure 10.** Proposed models of chromosome movements in meiosis I. (**A**) Proposed tug-of-war model for the initiation of X chromosome segregation in anaphase I in males. The X chromosome is shown in red, the holocentric kinetochore in gray, and the X chromosome-associated kinetochore microtubules in green. Left panel: At metaphase I, pulling forces at the X chromosome are in balance. Right panel: segregation of the X chromosome is initiated by an imbalance of forces, obvious by an unequal number of kinetochore microtubules associated with the opposite sides of the X chromosome. In anaphase I, the X is proposed to move to the side with more attached kinetochore microtubules. (**B**) Model illustrating the changes in spindle geometry. Left upper panel is metaphase; right upper panel is anaphase; lower panel shows combining anaphase A and B. Chromosomes are shown in blue, centrosomes in green, centrioles in black. Laterally associated microtubules are illustrated in orange. The end-on associated kinetochore microtubules (magenta in metaphase and green in anaphase) have the same length at both stages. The lower panel is an overlay of metaphase (magenta) with anaphase A (green, anaphase B movement was not considered) to show the relative movement of the autosomes with respect to the centrosomes. A simultaneous rounding of the autosomes, a shrinking of

the volume of the centrosomes and a change in the attachment angle of the microtubules is illustrated.

