## [Decision Letter]

**Acceptance summary:**

This study provides and important and beautiful description of the *C. elegans* spermatocyte spindle, which exhibits interesting and unexpected features distinct from many other mitoses. Overall, the reviewers agreed that this study provides important insights into *C. elegans* male meiotic divisions, laying the foundation for future studies.

**Decision letter after peer review:**

Thank you for submitting your article "Male meiotic spindle features that efficiently segregate paired and lagging chromosomes" for consideration by *eLife*. Your article has been reviewed by three peer reviewers, and the evaluation has been overseen by a Reviewing Editor and Anna Akhmanova as the Senior Editor. The reviewers have opted to remain anonymous.

The reviewers have discussed the reviews with one another and the Reviewing Editor has drafted this decision to help you prepare a revised submission.

Essential revisions:

1) All reviewers agreed that the writing and structure of the manuscript need substantial revisions. In its current form, the manuscript lacks clarity and accuracy. Reviewers discussed that the manuscript may be improved by re-organization (suggested by reviewer #2), and more careful writing in a number of places (individual points are listed below).

2) Reviewer #3 indicated that the conclusions regarding anaphase A are not well-supported by evidence and highlighted that the segregation of the X is not being considered by the authors when suggesting that k-fiber shortening does not drive chromosome-to-pole movement in these spindles. After discussion among reviewers, the important points raised by reviewer #3 must be taken into account when presenting analysis and conclusions related to anaphase A.

During our discussion, reviewer #2 suggested the following which may help partially address reviewer #3's comments and potentially amend some of the conclusions presented.

The authors show plots of what they refer to as end-on and lateral (see comments on these terms in the detailed reviews) for the X – however, all their plots would benefit from including the X-to-pole (as well as autosome-to-pole) separation distances to help interpret what is currently only referred to as anaphase 1,2,3, 4 in the graphs. At least in the plot shown it does not look like anaphase 4 (most separated) was associated with significantly shorter MT lengths but they should address reviewer #3's comments on this point.

The authors should also separate the two spindle halves (the "winning" vs. "losing" side) in the later anaphase stages and plot lengths as a function of X-pole distance in each half-spindle; this would help in terms of assessing whether X segregation was associated with significant shortening of microtubules that they detect by tomography. Currently all these values are averaged and, as noted by reviewer #3, the modest changes and large deviations make the conclusions stated about lack of microtubule shortening questionable.

3) Reviewers recommend that the authors conduct additional live imaging analysis of anaphase B (e.g. by visualizing central spindle components and cortical force generators), as that is the dominant mechanism separating chromosomes. Any insights from the tomography analysis on anaphase B should also be discussed to provide a more balanced view of the chromosome separation mechanism. Although live imaging would be ideal, if it is not feasible, the authors can characterize the localization of central spindle proteins (SPD-1, centralspindlin, etc.) by immunofluorescence, to better demonstrate that the central spindle really doesn't exist.

Overall, while significant additional experimental analysis is not necessary, we would like authors to: (1) reorganize the writing for clarity and accuracy; (2) re-analyze existing data (or potentially add new data if available to strengthen their points) and be significantly more careful about claims related to the minor anaphase A-like movement in light of the critiques from reviewer #3; and (3) present additional characterization of the predominant chromosome separation mechanism: anaphase B.

Reviewer #1:

The manuscript by Fabig et al. describes how chromosomes are segregated on the meiosis I (and II) spindles in *C. elegans* spermatocytes. Using a combination of live cell imaging, electron tomography and fixed images, the authors present a beautiful description of the spermatocyte spindle. This analysis reveals some very interesting and unexpected features including novel mechanisms to account for anaphase A movement, as well as providing insight into how lagging chromosomes are segregated.

In the Introduction, the authors conflate general features of chromosome segregation with the specifics of *C. elegans*. For example, Introduction paragraph two, they generalize findings in *C. elegans* that may or may not be the same in other systems. The authors need to be more careful in their description between what is happening in *C. elegans* versus general features of chromosome segregation.

Subsection “Lagging of chromosomes is a consequence of a lack of pairing” paragraph three: ZIM-2 mediates the pairing/synapsis of chromosome V not IV (Phillips and Dernburg, 2006).

Subsection “Spermatocyte meiotic centrosome and spindle dynamics are distinct from that in mitosis”: What is the evidence that spindle MTs remain connected to the X chromosome?

Figure 6: Please add the color code directly to the figure (e.g., white is end-on MT interactions and gold is lateral MT interactions). Would it be possible to highlight a continuous MT from the centrosome to the X chromosome?

Appendix—figure 7: Absence of central spindle. The authors present evidence that there is no central spindle based on weak MT staining in the center (and the presence of the lagging X chromosome). This would be strengthened by staining with central spindle components such as CYK-4/ZEN-4/SPD-1 in a similar manner as was done for the inner and outer kinetochore (Figure 4).

Is there any correlation between the curved MTs and which pole the X segregates to? In the images/videos the curve is more pronounced on one side of the spindle.

Reviewer #2:

The manuscript from Fabig et al. presents live imaging analysis of male meiotic divisions in *C. elegans* that is complemented by electron tomographic analysis of microtubule distributions during this developmentally specialized division. This is a technically strong study with two major themes: (1) analysis of the unpaired X chromosome's segregation – lagging followed by eventual segregation to one spindle pole, and (2) description of a modest anaphase A-like movement of autosomes that is not associated with microtubule shortening but instead driven by shape changes.

The manuscript has compelling elements but is a difficult read and would greatly benefit from reorganization and more careful writing. A more challenging question is whether there is need for any type of perturbation analysis – this point can be addressed in the reviewer discussion. To enhance the manuscript's impact, I would recommend the authors organize the manuscript into two sections – the first focused on X chromosome lagging and the second briefer one on anaphase A-like autosome movement – that are bridged by the tomography (which is the most important contribution here and needs to be presented significantly earlier in the paper). Here is a recommended structure along with comments on each section:

The authors need to make it significantly easier for readers to evaluate figures and compare different datasets. All time-dependent graphs should have a common time 0, e.g. anaphase onset, and the Time axis should be labeled "Time relative to anaphase onset (s)". On 2-color images shown, the component visualized in each color should be labeled on the figure and not only in the legend. Similarly, pseudo-colored entities in the tomograms should be described on the figure with proper labels. These types of modifications will great improve accessibility of the manuscript to interested readers.

1) Merge Figure 1 and 2 into one figure showing that unpaired chromosomes lag.

The authors state that "sister chromatids of the unpaired X attach to opposite poles…"; this is not in fact supported by the data and should be removed. An interesting point that should be mentioned is that cohesion is protected between the sisters of the X in meiosis I – presumably because the lack of recombination prevented definition of an axis of cohesion removal in meiosis I.

2) Leave Figure 3 for later in the paper (see below); move Figure 4 to the supplement – this figure overlaps with prior fixed data, does not report any significant new observations and is disruptive in the flow of the manuscript.

One point here – that HIM-10 looks different from MIS12 complex and KNL-1 – is rather surprising and should be verified using CRISPR-tagged versions that have been generated for all of these components. It is possible that the staining observed with the anti-HIM-10 antibody between the homologs is non-specific.

3) Make current Figure 5 into Figure 2. This figure is focused on movement of the lagging X and will follow directly from the revised Figure 1 (see point 1 above).

As noted above, label imaged components on the figure. There is one issue with the conclusions in Figure 1A and B – the change in chromosome morphology cannot be solely attributed to loss of tension because there is also reduction in cell cycle phosphorylation between meiosis I and meiosis II. The authors should analyze their laser ablation data (especially sequences like no. 2) to assess if there is rapid relaxation after ablation. In the absence of such d ata, they need to be more cautious in the interpretation in current Figure 5A and B.

4) Make current Figure 6 (tomography) into Figure 3 – label colors on the figure and not in legend. Provide numerical summary also in the figure (can be repeated in the text).

The last column is very difficult to visualize. On a more important note, the exact criterion used to call end-on vs. lateral should be clarified here even if they were previously described. Precisely how much length of microtubule should be present in the "clear" zone for it to be designated "lateral"? This is also a potential point of confusion in that the classification does not match what these terms are used to refer to in analysis in other systems (e.g. yeast or human cells, where a lateral attachment refers to kinetochore bound on the side of a microtubule that extends far past the kinetochore). I suspect that the attachments described here may be similar to what is termed an "end-on" attachment in other systems. There is also a geometric issue here coming from the curvature of the chromosomes – the surface that is available for what is classified as an end-on attachment may limit their number.

5) Make current Figure 7 and Figure 8 into Figures 4 and 5. These continue to focus on the lagging X chromosome and its segregation but now integrate the tomography.

There are some issues with the writing and figure elements associated with this analysis. In the tortuosity analysis, is there any contribution from the cleavage furrow? Is there a furrow at the stages that are visualized or not? In current Figure 8B, what are "anaphase S1, S2, S3"? In Figure 8F, why does the ratio go up? Is it because more microtubules are formed on the "winning" side or there is a decrease in microtubules on the "losing" side? This should be clarified by looking at the measured values and not the ratio. More generally, there is not any causal relationship established by any of this analysis and the writing needs to be more circumspect. There is also an analogy to made to segregation of merotelically attached kinetochores in mitotic mammalian cells that lag and segregate to the side that ends up with more microtubules. This should be mentioned.

6) End the paper with 2 figures (Figure 6 and 7) on autosomal anaphase A (which provides ~20% of separation). Start Figure 6 with current Figure 9 to highlight that kinetochore microtubules on autosomes did not shorten and follow that with current Figure 2, based on light imaging, which shows there is modest anaphase A, as defined by reduction in distance of chromosomes to poles. Then end with what is currently Figure 10, which provides reasons for why this would be the case.

Note that the chromosome shape change and the angular change are related and not independent – thus there are 2 factors at play – the chromosome shape change and the change in the centrosome. It would be helpful if the Anaphase 1, 2, 3 and 4 were annotated to include the separation distance of the autosomes.

On a related note, 80% or more of the separation is not due to anaphase A. Is this entirely due to cortical pulling? Given their expertise, the authors should look at conserved central spindle markers (SPD-1, CYK-4, ZEN-4) and also dynein (DHC-1) – there should be endogenously tagged versions of all of these components available by now.

Reviewer #3:

This manuscript by Fabig et al. reports a detailed characterization of spermatocyte meiosis in *C. elegans*. The authors present data that provides some new insights into spindle organization and chromosome segregation in this system. However, I find some of their conclusions to be insufficiently supported (see points below).

1) One of the major points that this paper attempts to make is that k-fiber shortening does not drive anaphase A in spermatocytes. However, I do not think that the presented data provide strong support for this conclusion. One of the main issues is that the spermatocyte spindles are small and the poleward-facing edges of the autosomes start out very close to the poles, so the chromosomes do not move very far in anaphase A. Therefore, if there was k-fiber shortening it would be hard to detect, especially given the variability in microtubule lengths reported from the electron tomography analysis. The authors report that the "end-on" microtubules for metaphase are 0.62 +/- 0.33 µm and the "lateral" microtubules are 1.15 +/- 0.59 µm. Given this amount of variability, if some population of these microtubules shortened and helped drive movement towards the pole, it could be hard to detect. Compounding this issue, since only one metaphase spindle was analyzed, it is difficult to know whether the numbers reported for metaphase are consistent from spindle to spindle.

Additionally, the author's own data appears to suggest that k-fiber shortening could be capable of driving poleward movement in these spindles. In the case of the lagging X chromosome, the authors show nice videos of the X segregating in late anaphase (both in the normal case in Figure 1/Video 1, and in their cutting experiments in Figure 5/Video 4). In these cases (since the spindle has elongated in anaphase B), the X has to segregate over a longer distance and therefore it is easier to visualize what is happening to the microtubules/k-fibers. In these videos, it appears that once the X starts segregating to the "winning" pole, the corresponding k-fiber shortens, reeling in the chromosome. It seems unlikely to me that this k-fiber shortening mechanism to drive chromosome-to-pole movement would exist for the X chromosome, but not the autosomes. It seems more likely that the autosomes also segregate by this mechanism in anaphase A, but that it is difficult to measure because the distances are so short and the variability of the microtubule lengths is so high. Therefore, I do not find the conclusions of the authors about anaphase A mechanisms to be convincing.

2) The designation of "end-on" vs. "lateral" microtubules in Figure 6 is confusing. Many of the white microtubules designated as end-on appear to run quite far down the side of the chromosome. Therefore, they don't really appear to be "end-on". Correctly categorizing microtubules is important, because the authors make a major point about end-on microtubules not shortening during anaphase, and use this as evidence to say that k-fiber shortening does not drive anaphase A movements. But if some of the microtubules they are counting are actually running along the sides of the chromosomes and if this population of lateral microtubules do not shorten, then the authors may be missing a reduction in the length of the ones that actually are "end-on".

Related to this, in paragraph four of subsection “Tension release across the spindle may contribute to autosomal anaphase A”, the authors use their conclusion that microtubule lengths are constant at 0.63 µm between metaphase and anaphase to calculate how much altering the microtubule angle could contribute to chromosome movements (ending up with a value of 0.17 µm). However, since this measurement of microtubule length is subject to error (given the variability of microtubule length measurements and the potential issue with distinguishing end-on from lateral interactions), the calculations that led to the 0.17 µm shortening number are also in question.

3) In Figure 10, the authors report the ANOVA comparing metaphase with two of the anaphase spindles (number 3 and 4) and it appears that there is a significant difference, but they do not report the ANOVA comparison between metaphase and anaphase numbers 1 and 2 (and #2 especially does not appear to be different from metaphase). This raises the concern that variability between spindles could account for the different amounts of autosomal stretching, rather than spindle stage (metaphase vs. anaphase). This is particularly important since the authors use this data to propose that release of this stretch can account for much of the "anaphase A" pole-chromosome shortening. To me, it seems problematic to draw these conclusions when only one metaphase spindle is analyzed, and therefore it is difficult to exclude spindle-spindle differences in the autosomal FWHMs.

4) For the analysis of chromosome segregation, it would be helpful to have more information about how the distances were determined. The Experimental Procedures state "For the analysis of chromosome movements, the peak maxima of the chromosome fluorescence signals were then used to calculate the distances for each time point." This is confusing and more details on the analysis should be included so the reader can better evaluate these data – how were the centers of each autosome determined? I can imagine that this would be difficult given the resolution of the videos, and subject to error. Also, was each autosome within a given spindle measured separately, or was each segregating mass of autosomes treated as one unit (and the center of that entire mass determined)? Given potential issues in accurately determining the "center" (of either each autosome or the autosome mass), I would suggest that the authors try measuring the distance between the outer edges of the autosomes (the poleward facing sides) and the spindle pole… this would more clearly represent the distance the chromosome travels towards the spindle pole.

5) The authors state that: "Interestingly, the centrosomes remain connected to the X chromosome-connected microtubules". However, there is no figure call-out for this statement, and I can't find any figures where there is convincing evidence of this, given the resolution of the images/videos presented. I am therefore confused as to what evidence supports the subsequent statement "Thus, the separation and migration of centrosomes during anaphase I appears to be coordinated with microtubules that must maintain connections not only to segregating autosomes, but also the lagging X chromosome". Either present these data or revise these statements.

---

## [Author Response]

Essential revisions:1) All reviewers agreed that the writing and structure of the manuscript need substantial revisions. In its current form, the manuscript lacks clarity and accuracy. Reviewers discussed that the manuscript may be improved by re-organization (suggested by reviewer #2), and more careful writing in a number of places (individual points are listed below).

We thank all of the reviewers for their insightful suggestions and comments. We have substantially restructured the entire manuscript mainly adhering to suggestions for reorganization by reviewer #2. In addition, we collected 2 new tomographic data sets and reanalyzed or reorganized the original tomographic data to address reviewer concerns. As suggested, we also include additional immunostaining and live imaging data to be clearer about the structure of inter-chromosomal microtubules and the extent of anaphase A (see also below). Despite the addition of new data, we streamlined the whole text as suggested by the reviewers. Importantly, we modified our conclusions to take into account all suggestions and concerns.

2) Reviewer #3 indicated that the conclusions regarding anaphase A are not well-supported by evidence and highlighted that the segregation of the X is not being considered by the authors when suggesting that k-fiber shortening does not drive chromosome-to-pole movement in these spindles. After discussion among reviewers, the important points raised by reviewer #3 must be taken into account when presenting analysis and conclusions related to anaphase A.During our discussion, reviewer #2 suggested the following which may help partially address reviewer #3's comments and potentially amend some of the conclusions presented.The authors show plots of what they refer to as end-on and lateral (see comments on these terms in the detailed reviews) for the X – however, all their plots would benefit from including the X-to-pole (as well as autosome-to-pole) separation distances to help interpret what is currently only referred to as anaphase 1,2,3, 4 in the graphs. At least in the plot shown it does not look like anaphase 4 (most separated) was associated with significantly shorter MT lengths but they should address reviewer #3's comments on this point.

We have made several changes to address the concerns of reviewer #3 in regard to our conclusions about anaphase A. These are described in detail below in our responses to individual reviewer comments. Our intent is not to say that there is a complete absence of kinetochore microtubule shortening but instead to show that there are alternative mechanisms that we identified that can contribute to anaphase A.

To address the suggested changes by reviewer #2, we added additional analysis to Figure 5. From what we could interpret from the suggestion, we added an additional analysis to Figure 5, which shows the microtubule length of either side of the X chromosome plotted against the respective X chromosome-to-pole distance. In Figure 5C it is possible now to distinguish the microtubule length of either side of the X chromosome in relation to the X chromosome-to-pole distance. This illustrates that for the majority of data sets the closer pole to X has also shorter microtubules and vice versa.

We also provide more detail and clarity about our staging of the reconstructed spindles. We compared our tomographic data with live-imaging data of chromosome and pole dynamics (see Appendix—figure 2). In order to keep our plots ‘readable’, we omitted the addition of the suggested X-to-pole distance or autosome-to-pole distance to each figure. All this data is also summarized in Table 1 and plotted in Figure 5D and in the additional Appendix—figure 9.

The authors should also separate the two spindle halves (the "winning" vs. "losing" side) in the later anaphase stages and plot lengths as a function of X-pole distance in each half-spindle; this would help in terms of assessing whether X segregation was associated with significant shortening of microtubules that they detect by tomography. Currently all these values are averaged and, as noted by reviewer #3, the modest changes and large deviations make the conclusions stated about lack of microtubule shortening questionable. It is important that the authors take these comments seriously when preparing their revision.

Our electron tomograms are snapshots of specific stages of meiotic divisions. Thus, assigning labels like “winning” and “losing” to each side of the single X would be speculative, which we wish to avoid. An unbiased way of analysis is to plot the mean microtubule length of microtubule from each side of the X chromosome against its X chromosome distance. As suggested, we have done this in our new Figure 5C. There is a clear linear relationship between the distance of the X chromosome to the spindle pole and the microtubule length. We also created a second new plot (Figure 5D) that color codes the identical data by the type of microtubule association to the chromosome. This way of plotting the data now illustrates nicely that both end-on and laterally associated microtubules change their length in response to the distance of the X chromosome to the respective spindle pole with similar characteristics. Again, in light of reviewer #3’s comments, we have amended the text to adjust our conclusions regarding anaphase A as described above. We have amended the text in multiple places to make this clearer which is detailed in the response below.

3) Reviewers recommend that the authors conduct additional live imaging analysis of anaphase B (e.g. by visualizing central spindle components and cortical force generators), as that is the dominant mechanism separating chromosomes. Any insights from the tomography analysis on anaphase B should also be discussed to provide a more balanced view of the chromosome separation mechanism. Although live imaging would be ideal, if it is not feasible, the authors can characterize the localization of central spindle proteins (SPD-1, centralspindlin, etc) by immunofluorescence, to better demonstrate that the central spindle really doesn't exist.

As suggested we have amended the text to better describe our characterization of sperm-specific mid-zone spindle dynamics. Our intent is to show that these dynamics are distinct from those of oocyte meiosis or mitosis, not necessarily that the central spindle doesn’t exist. Our revised manuscript now includes added immunolocalization of proteins that specify the central spindle during anaphase I to show that, instead of localizing in the spindle midzone during mid-late anaphase I as in oocyte meiosis or mitosis, they localize either on chromosomes (Aurora BAIR-2 or CLASPCLS-2) or, strikingly, at the ingressing membrane (MKLP1ZEN-4). We have also included live-imaging analysis of the localization dynamics of GFP-tagged PRC1SPD-1 during sperm meiotic anaphase I, which shows it localizes to the midzone, but its presence decreases during anaphase I progression, which is unlike oocyte meiosis and mitosis when its levels persist throughout anaphase.

Overall, while significant additional experimental analysis is not necessary, we would like authors to: (1) reorganize the writing for clarity and accuracy; […]

We have largely reorganized the paper as suggested by reviewer #2.

[…] (2) re-analyze existing data (or potentially add new data if available to strengthen their points) and be significantly more careful about claims related to the minor anaphase A-like movement in light of the critiques from reviewer #3; […]

We did significant additional analysis (see below) to thoroughly address reviewer comments. In particular, we added more data and re-analyzed previously presented data related to the minor anaphase A-like movement, which we describe in the detailed responses to individual reviewer comments below. We have also amended our claims throughout to clarify our conclusions – in particular include the possibility that shortening of kinetochore microtubules may play some role, but in combination with the new factors we describe.

[…] and (3) present additional characterization of the predominant chromosome separation mechanism: anaphase B.

We provide additional data, including immunolocalization and live-imaging of central spindle specifiers, to support that midzone spindle structures during anaphase I are distinct in sperm meiosis compared to oocyte meiosis and mitosis. We have also amended our claims throughout to clarify our conclusion that there are alternate midzone spindle structure and dynamics during sperm meiosis that support a stronger reliance on non-midzone generated forces compared to oocyte meiosis and mitosis.

Reviewer #1:The manuscript by Fabig et al. describes how chromosomes are segregated on the meiosis I (and II) spindles in *C. elegans* spermatocytes. Using a combination of live cell imaging, electron tomography and fixed images, the authors present a beautiful description of the spermatocyte spindle. This analysis reveals some very interesting and unexpected features including novel mechanisms to account for anaphase A movement, as well as providing insight into how lagging chromosomes are segregated.

Thank you for this comment.

In the Introduction, the authors conflate general features of chromosome segregation with the specifics of *C. elegans*. For example, Introduction paragraph two, they generalize findings in *C. elegans* that may or may not be the same in other systems. The authors need to be more careful in their description between what is happening in C. elegans versus general features of chromosome segregation.

Thank you for pointing this out. In the revised version we have reorganized the Introduction to clarify features of chromosome segregation that pertain to *C. elegans* meiosis.

Subsection “Lagging of chromosomes is a consequence of a lack of pairing” paragraph three: ZIM-2 mediates the pairing/synapsis of chromosome V not IV (Phillips and Dernburg, 2006).

Thank you for pointing out this mistake in our manuscript. We have corrected this.

Subsection “Spermatocyte meiotic centrosome and spindle dynamics are distinct from that in mitosis”: What is the evidence that spindle MTs remain connected to the X chromosome?

It is traditionally hard to show, even by electron tomography, that microtubules are directly “connected’ to the holocentric *C. elegans* kinetochore, as the kinetochore itself is not visible as an electron-dense plaque like the kinetochore in mammalian cells. For this reason, we decided to be more cautious and change our wording to “associated’. To support the association of microtubule association to chromosomes, in the manuscript we refer here to microtubules which end or travers the ribosome-free zone around the chromosomes with at least 150 nm distance in *C. elegans* males (paragraph two subsection “Spermatocyte spindles maintain both end-on and lateral associations of

kinetochore microtubules to chromosomes throughout meiosis”), which is considered the holocentric kinetochore In addition to our EM data, which clearly shows that single microtubules bridge the distance between spindle poles and the X chromosome (Figure 5 and Videos 8-11), we also show in Figures 2E-F and Appendix—figure 1F that the outer kinetochore stays assembled during anaphase and is present in ~150nm distance between chromosomes (including the X) and microtubules.

Figure 6: Please add the color code directly to the figure (e.g., white is end-on MT interactions and gold is lateral MT interactions). Would it be possible to highlight a continuous MT from the centrosome to the X chromosome?

As suggested, we have added the color coding to the figures containing electron tomographic data (now Figures 3 and 4, Appendix—figure 4 and Appendix—figure 7). Actually, almost all microtubules are spanning the distance between the X and the spindle pole. For a better visualization of the connection to the poles, we modified our videos of the tomographic reconstructions (Videos 4-11) to highlight all X chromosome-associated microtubules (in cyan).

Appendix—figure 7: Absence of central spindle. The authors present evidence that there is no central spindle based on weak MT staining in the center (and the presence of the lagging X chromosome). This would be strengthened by staining with central spindle components such as CYK-4/ZEN-4/SPD-1 in a similar manner as was done for the inner and outer kinetochore (Figure 4).

As suggested, we now show additional data to strengthen the point that a “classic” central spindle with interdigitating microtubules during mid-anaphase was not observed in meiosis I in *C. elegans* males. First, by analyzing *tra-2* XX males, which do not have a lagging X chromosome, by electron tomography, we find the microtubules appear to emanate from the poles and do not form a classic overlap, which is distinct from that observed in early mitosis or female meiosis in *C. elegans* (subsection “Interdigitating midzone microtubules are not a prominent feature during spermatocyte

anaphase progression”, Appendix—figure 7C). Second, we added new immunostaining data to show that the Aurora B kinase AIR-2 and the microtubule stabilizer CLASPCLS-2 remain chromosome associated in both *him-8* (with a lagging X) and *tra-2* (without a lagging X) animals during sperm meiotic anaphase I, in contrast to their strong midzone localization during oocyte meiosis and mitosis (Appendix—figure 8A). The centralspindlin component MKLP1ZEN-4 also shifts to localization at the ingressing cell membrane during mid-anaphase I. Further, using live-imaging we find PRC1SPD-1 exhibits transient presence in the midzone then is absent even as chromosome continue segregating during anaphase I progression (Appendix—figure 8B). A comparison of the spindle dynamics in *tra-2* males with wild-type males showed that spindles without lagging chromosomes exhibit a faster pole-pole separation with a longer final spindle length (Appendix—figure 7B). This indicates that the microtubule bridge in wild-type males rather counteracts spindle elongation than promoting it. In summary, we present evidence that spermatocytes have alternate spindle structure and dynamics that do not resemble the classic spindle midzone observed in oocyte meiosis and embryonic mitosis.

Is there any correlation between the curved MTs and which pole the X segregates to? In the images/videos the curve is more pronounced on one side of the spindle.

We find that the curving of microtubules associated with the X is very interesting and also puzzling. At this stage, it is difficult to make strong statements as to which extent straight versuscurved microtubules contribute to the segregation of the X. However, we consider it important to report this phenomenon for further consideration and investigation (see Appendix—figure 6).

Reviewer #2:The manuscript from Fabig et al. presents live imaging analysis of male meiotic divisions in *C. elegans* that is complemented by electron tomographic analysis of microtubule distributions during this developmentally specialized division. This is a technically strong study with two major themes: (1) analysis of the unpaired X chromosome's segregation – lagging followed by eventual segregation to one spindle pole, and (2) description of a modest anaphase A-like movement of autosomes that is not associated with microtubule shortening but instead driven by shape changes.The manuscript has compelling elements but is a difficult read and would greatly benefit from reorganization and more careful writing. A more challenging question is whether there is need for any type of perturbation analysis – this point can be addressed in the reviewer discussion. To enhance the manuscript's impact, I would recommend the authors organize the manuscript into two sections – the first focused on X chromosome lagging and the second briefer one on anaphase A-like autosome movement – that are bridged by the tomography (which is the most important contribution here and needs to be presented significantly earlier in the paper). Here is a recommended structure along with comments on each section:

We are very grateful for the time and effort this reviewer made to make clear suggestions to improve the text. We have changed the structure largely as suggested.

The authors need to make it significantly easier for readers to evaluate figures and compare different datasets. All time-dependent graphs should have a common time 0, e.g. anaphase onset, and the Time axis should be labeled "Time relative to anaphase onset (s)".

We have made the suggested changes to the figures as suggested to address this concern. Now, time (t=0) always refers to the onset of anaphase throughout the paper, with two exceptions (Figure 2B and Figure 6F). For these two plots we used “Time relative to the onset of X segregation”. The reasons for these two exceptions are as follows: The start of X-chromosome movement varied considerably from spindle to spindle, from 1.5–8.5 minutes after anaphase onset (mean: 4.92 +/- 1.55 min). Because of this variability, we decided after careful consideration to plot the shape coefficient relative to the onset of X chromosome movement. The point of Figure 2B is to clearly indicate the change in shape of the X chromosome in response to the onset of its movement. We decided to keep the data in Figure 6F (ratio of microtubules on both sides of the X chromosome) aligned to the onset of X chromosome movement for the same reason. We tried to make this difference to the other plots of spindle dynamics clearer by relabeling the X-axis of both graphs.

On 2-color images shown, the component visualized in each color should be labeled on the figure and not only in the legend. Similarly, pseudo-colored entities in the tomograms should be described on the figure with proper labels. These types of modifications will great improve accessibility of the manuscript to interested readers.

We agree and have added legends with color coding to the individual figures (Figures 1, 2, 6 and 7) and videos (in addition to the information that is given in the figure legends).

1) Merge Figure 1 and 2 into one figure showing that unpaired chromosomes lag.The authors state that "sister chromatids of the unpaired X attach to opposite poles…"; this is not in fact supported by the data and should be removed. An interesting point that should be mentioned is that cohesion is protected between the sisters of the X in meiosis I – presumably because the lack of recombination prevented definition of an axis of cohesion removal in meiosis I.

We have merged Figures 1 and 2 as suggested and removed the statement about the attachment of the sister chromatids. As for the second point, we feel that this information may also lack sufficient support by our data, thus we did not include it.

2) Leave Figure 3 for later in the paper (see below); move Figure 4 to the supplement – this figure overlaps with prior fixed data, does not report any significant new observations and is disruptive in the flow of the manuscript.One point here – that HIM-10 looks different from MIS12 complex and KNL-1 – is rather surprising and should be verified using CRISPR-tagged versions that have been generated for all of these components. It is possible that the staining observed with the anti-HIM-10 antibody between the homologs is non-specific.

We largely made the suggested changes. We removed the previous Figure 4 in large parts. We also amended our text to de-emphasize distinctions in specific kinetochore protein localization. The point we would like to make is that outer kinetochore proteins (such as KNL-1, KNL-3, NDC-80) are retained on meiotic chromosomes, including X, during anaphase I, which is distinct from what occurs in oocyte meiosis (now Figure 2E). This has previously not been reported and supports that kinetochore proteins bridge chromosome-microtubule interactions during anaphase I as chromosomes are pulled apart, and important for the segregation of the X chromosome.

3) Make current Figure 5 into Figure 2. This figure is focused on movement of the lagging X and will follow directly from the revised Figure 1 (see point 1 above).As noted above, label imaged components on the figure. There is one issue with the conclusions in Figure 1A and B – the change in chromosome morphology cannot be solely attributed to loss of tension because there is also reduction in cell cycle phosphorylation between meiosis I and meiosis II. The authors should analyze their laser ablation data (especially sequences like no. 2) to assess if there is rapid relaxation after ablation. In the absence of such data, they need to be more cautious in the interpretation in current Figure 5A and B.

The sequence of the figures and information on the color coding has been added as suggested. As for the relaxation analysis, we agree that this is an issue that needs to be taken in account. For the laser ablation we did not use 3D imaging but instead imaged a single plane with manual focusing, but with a high time resolution. Only in this way was it possible to ablate an object in the focal plane reliably. As such, we cannot analyze the shape of the X chromosome in a similar way as shown in Figure 2A and B. It was also challenging to keep the cells in focus due to the movements of the imaged worms. We also had to deal with slightly tilted random orientations of the meiotic spindles. For these technical reasons, we could not analyze the shape of the X chromosome after laser ablation, although we agree that such an analysis would be informative.

It is not clear to us how to address the comment on how reduction in cell cycle phosphorylation might change chromosome morphology between meiosis I and II. We do admit there might be multiple factors involved, though loss of tension seemed to be the most obvious contributor. Thus, due to time and space restrictions, cell cycle phosphorylation could not be analyzed or included within this study.

4) Make current Figure 6 (tomography) into Figure 3 – label colors on the figure and not in legend. Provide numerical summary also in the figure (can be repeated in the text).The last column is very difficult to visualize. On a more important note, the exact criterion used to call end-on vs. lateral should be clarified here even if they were previously described. Precisely how much length of microtubule should be present in the "clear" zone for it to be designated "lateral"? This is also a potential point of confusion in that the classification does not match what these terms are used to refer to in analysis in other systems (e.g. yeast or human cells, where a lateral attachment refers to kinetochore bound on the side of a microtubule that extends far past the kinetochore). I suspect that the attachments described here may be similar to what is termed an "end-on" attachment in other systems. There is also a geometric issue here coming from the curvature of the chromosomes – the surface that is available for what is classified as an end-on attachment may limit their number.

We made the suggested figure change.

We are aware of the “geometric issue “that has been raised here. Indeed, end-on versuslateral association of KMTs with chromosomes is clearly defined in systems with monocentric kinetochores (for example, in budding yeast with one kinetochore microtubule per chromosome). As published for previous EM studies in *C. elegans* (for instance in Howe et al., 2001; O’Toole et al., 2003; Redemann et al., 2017), microtubules that are submerged in a ribosome-clear zone (representing the holocentric or diffuse kinetochore) were considered as kinetochore microtubules. We did not define a minimal distance in our analysis. Indeed, the situation is complicated by the fact that chromosomes in *C. elegans* meiosis are not rod-like but cup-shaded. Because of this difference in geometry, the probability of a lateral interaction of a microtubule with a chromosome is increased. In addition, and this is a very important point, the tomographic data show that the chromosome surface is not smooth but instead rather ‘rough’ or ‘curvy’. This makes the analysis even more complicated. All these considerations were taken into account when developing our approach of defining an end-on versusa lateral association. As described in detail in Materials and Methods, we modeled the chromosome surface. Then we addressed each microtubule that was at least 150 nm or closer with any part of its lattice to the surface of the chromosomes. Next, we extrapolated all microtubule ends by 150 nm and scanned for a crossing of the chromosome surface and defined the positive one as end-on associated. All other microtubules, not crossing the chromosome surface that still were 150 nm or closer next to the chromosomes were defined as laterally associated. To further illustrate this method, we added a video (see new Video 15, called out in Materials and Methods) showing both types of microtubule association to chromosomes in one 3D model at high resolution. To our mind, the developed approach is a clear und unbiased way to analyze the tomographic data. We agree that one should be more careful here in making strong conclusions. Thus, we simply want to report that both end-on and lateral associations of microtubules to chromosomes exist throughout male meiosis. We avoid strong conclusions about the function of end-on versuslateral associations as discussed in the literature for microtubules attached to a monocentric kinetochore. This issue needs to be addressed in a follow-up paper.

5) Make current Figure 7 and Figure 8 into Figures 4 and 5. These continue to focus on the lagging X chromosome and its segregation but now integrate the tomography.There are some issues with the writing and figure elements associated with this analysis. In the tortuosity analysis, is there any contribution from the cleavage furrow? Is there a furrow at the stages that are visualized or not?

The figures have been changed as suggested.

As for the role/contribution of the ingressing furrow, we did some experiments to investigate whether the membrane plays a role here (data not shown in the paper). From the videos we recorded it was our impression that the decision in terms of direction was done before the ingressing membrane has a restrictive effect on the segregation, thus we are unable to conclude it has a clear role. In our tomograms we do not see the ingressing membrane. Even our most advanced stage (anaphase no. 7) does not show the cleavage furrow. We must collect much more data to be definitive about that issue, which we feel would be more appropriate in a follow up paper.

In current Figure 8B, what are "anaphase S1, S2, S3"? In Figure 8F, why does the ratio go up? Is it because more microtubules are formed on the "winning" side or there is a decrease in microtubules on the "losing" side? This should be clarified by looking at the measured values and not the ratio. More generally, there is not any causal relationship established by any of this analysis and the writing needs to be more circumspect.

We have clarified the description of all the data we analyzed and included these data sets to make the information available to other researchers. The data sets Anaphase S1-3 of the previous submission are additional reconstructions shown now as Appendix Material (Appendix—figure 4). We have also renamed theses data sets to anaphase no. 2, no. 5 and no. 6, to make their stage of division more clear (see Figure 6D).

We also clarified the presentation of data regarding resolution of the X chromosome. For the tomographic data, in the new Figure 6B we give the ratio of microtubule content and in Figure 6D we show the microtubule numbers for each 3D reconstruction. Indeed, the ratio indicates that one side has more microtubules, while the other one shows less microtubules. Though it would be helpful to give numbers for the “winning” and the “losing” side, we would like to point out that this is not possible for early to mid-anaphase stages by looking at the static 3D models to know which direction the X would have migrated. We were thus purposefully careful to provide just the ratio of the numbers without mentioning the direction for the tomographic data. However, to address this we provide quantification of live-imaging data to support the level of connections to the “winning” and “losing” side to further support our findings (subsection “Segregation of the X chromosome correlates with an asymmetry in the number of associated microtubules”). As shown in Figure 6E-F, we see an increase in the fluorescence intensity on one side of the X and a decrease in the intensity on the other side of the X chromosome. As a general comment, we take care in the manuscript to say the data is supportive of a model for the decision related to the direction of the X chromosome segregation. We also indicate more data sets of very late stages must be analyzed. Because each tomographic reconstruction is a tremendous amount of work, this will certainly be another “tour de force “study and we have planned already to start this kind of analysis.

There is also an analogy to made to segregation of merotelically attached kinetochores in mitotic mammalian cells that lag and segregate to the side that ends up with more microtubules. This should be mentioned.

Thank you for pointing this out. We have added a comment to address this in the Discussion.

6) End the paper with 2 figures (Figure 6 and 7) on autosomal anaphase A (which provides ~20% of separation). Start Figure 6 with current Figure 9 to highlight that kinetochore microtubules on autosomes did not shorten and follow that with current Figure 2, based on light imaging, which shows there is modest anaphase A, as defined by reduction in distance of chromosomes to poles. Then end with what is currently Figure 10, which provides reasons for why this would be the case.Note that the chromosome shape change and the angular change are related and not independent – thus there are 2 factors at play – the chromosome shape change and the change in the centrosome. It would be helpful if the Anaphase 1, 2, 3 and 4 were annotated to include the separation distance of the autosomes.

We have changed the sequence of the figures as suggested. We have also added the requested information about the autosome-to-autosome distances in Figures 3-4 and Appendix—figure 4. This information is also given in Table 1.

On a related note, 80% or more of the separation is not due to anaphase A. Is this entirely due to cortical pulling? Given their expertise, the authors should look at conserved central spindle markers (SPD-1, CYK-4, ZEN-4) and also dynein (DHC-1) – there should be endogenously tagged versions of all of these components available by now.

Related also to comments from reviewer #3, we provide additional immunostaining data and live imaging to address this comment (subsection “Interdigitating midzone microtubules are not a prominent feature during spermatocyte anaphase progression”, Appendix—figure 7). In contrast to oocyte meiosis and mitosis, during sperm meiotic anaphase I Aurora B kinase AIR-2 and the microtubule stabilizer CLASPCLS-2 remain chromosome associated in both *him-8* (with a lagging X) and *tra-2* (without a lagging X) animals. Further, PRC1SPD-1 exhibits decreasing presence in the midzone during anaphase I progression while the centralspindlin component MKLP1ZEN-4 shifts to localization at the ingressing cell membrane during mid-anaphase I. These results support that spermatocytes have sperm-specific spindle structure and dynamics. This further supports a reliance on cortical forces, instead of a microtubule pushing as observed in oocyte meiosis and also in mitosis after laser microsurgery. We believe a detailed analysis of the role of cortical pulling mechanisms should be part of a further study.

Reviewer #3:This manuscript by Fabig et al. reports a detailed characterization of spermatocyte meiosis in *C. elegans*. The authors present data that provides some new insights into spindle organization and chromosome segregation in this system. However, I find some of their conclusions to be insufficiently supported (see points below).1) One of the major points that this paper attempts to make is that k-fiber shortening does not drive anaphase A in spermatocytes. However, I do not think that the presented data provide strong support for this conclusion. One of the main issues is that the spermatocyte spindles are small and the poleward-facing edges of the autosomes start out very close to the poles, so the chromosomes do not move very far in anaphase A. Therefore, if there was k-fiber shortening it would be hard to detect, especially given the variability in microtubule lengths reported from the electron tomography analysis. The authors report that the "end-on" microtubules for metaphase are 0.62 +/- 0.33 µm and the "lateral" microtubules are 1.15 +/- 0.59 µm. Given this amount of variability, if some population of these microtubules shortened and helped drive movement towards the pole, it could be hard to detect. Compounding this issue, since only one metaphase spindle was analyzed, it is difficult to know whether the numbers reported for metaphase are consistent from spindle to spindle.

We understand the reviewer’s issue with the strong conclusion that kinetochore microtubule shortening has no role in anaphase A that we detect in sperm meiosis. We were also surprised by the fact that one can see a clear anaphase A movement by light microscopy but not a clear shortening of kinetochore microtubules by electron tomography reaching single-microtubule resolution. From our analysis of the data, we believe that the spread of the data is due to the cup-shape and the ‘curvy surface’ of the autosomes (see Video 15). We also acknowledge that given this spread of the data that the reviewer points out, we cannot exclude the possibility that there could be some shortening of a subfraction of microtubules. Nonetheless, the degree to which we find that the majority of microtubules do not shorten to account for the total anaphase A distance traveled is still an interesting observation. To address the valid concerns raised by reviewer #3, we have restructured the paper and amended the text to not preclude that shortening of kinetochore microtubules may contribute in some degree to anaphase A. For example: “…our tomographic analysis suggests that shortening of kinetochore microtubules does not fully account for the anaphase A observed by light microscopy.” Also, in the Discussion we state: “Microtubules can shorten during anaphase I, as observed when the X resolves to one side; thus, a shortening of a subset of microtubules that is difficult to detect by current methods may also contribute to a small portion of anaphase A movement. Nonetheless, our proposed new mechanisms can now be considered when analyzing anaphase A movement in other systems.”

We agree with the reviewer that more metaphase data would strengthen our conclusions. To address this, we recorded and analyzed two more metaphase I tomographic data sets. These additional full data sets are now included in the manuscript. One of the new data sets (metaphase data set no. 3) is in full agreement with the already presented data. The second additional data set (metaphase no. 1), however, is slightly different. Chromosomes don’t show the typical stretching of the chromosomes known to occur at peak of metaphase immediately before anaphase onset. In addition, the pole-to-pole distance and the autosome-to-autosome distance is smaller compared to the other two data sets. For these reasons, we consider this data to represent a spindle at the prometaphase to metaphase transition or a very early metaphase.

In summary for the raised point, the conflict here remains that light microscopy shows a clear anaphase A movement (i.e. a contribution to the segregation of about 20%), whereas electron microscopy does not show a significant shortening of a large portion of the autosome-associated kinetochore microtubules, which we believe is an interesting fact. At this point, it is important to present the length measurements in a clear way and it is important to consider these values in the context of other changes in the cellular ultrastructure, of which we hope we now clearly describe and quantify to characterize other factors that can account or anaphase A displacement.

Additionally, the author's own data appears to suggest that k-fiber shortening could be capable of driving poleward movement in these spindles. In the case of the lagging X chromosome, the authors show nice videos of the X segregating in late anaphase (both in the normal case in Figure 1/Video 1, and in their cutting experiments in Figure 5/Video 4). In these cases (since the spindle has elongated in anaphase B), the X has to segregate over a longer distance and therefore it is easier to visualize what is happening to the microtubules/k-fibers. In these videos, it appears that once the X starts segregating to the "winning" pole, the corresponding k-fiber shortens, reeling in the chromosome. It seems unlikely to me that this k-fiber shortening mechanism to drive chromosome-to-pole movement would exist for the X chromosome, but not the autosomes. It seems more likely that the autosomes also segregate by this mechanism in anaphase A, but that it is difficult to measure because the distances are so short and the variability of the microtubule lengths is so high. Therefore, I do not find the conclusions of the authors about anaphase A mechanisms to be convincing.

It was not our intention to suggest that there is no kinetochore microtubule shortening possible during sperm meiosis, particularly because we did observe shortening of X-associated microtubules during later stages of anaphase I (Figure 5C). We cannot preclude that kinetochore microtubules connected to autosomes may shorten to some extent, though we provide quantitative measurements it is unlikely that this possible shortening could account for the full anaphase A displacement we find. In addition, we observe that X chromosome segregation takes place after autosomes have been partitioned to the opposite poles. During early anaphase when autosomes are separating, we actually see X-connected microtubules elongating, not shortening. We have amended the text to better reflect that we think the most likely explanation is that microtubules can grow and shorten very dynamically and depend on the balance of forces experienced by chromosomes at the time. In our revision, we simply provide evidence that the release of tension at anaphase I onset has the additional factors of tension release that contribute to anaphase A movement, not that kinetochore shortening does not occur at all. We have also amended the text to better indicate that anaphase A contributes only ~20% to the segregation of autosomes.

2) The designation of "end-on" vs. "lateral" microtubules in Figure 6 is confusing. Many of the white microtubules designated as end-on appear to run quite far down the side of the chromosome. Therefore, they don't really appear to be "end-on". Correctly categorizing microtubules is important, because the authors make a major point about end-on microtubules not shortening during anaphase, and use this as evidence to say that k-fiber shortening does not drive anaphase A movements. But if some of the microtubules they are counting are actually running along the sides of the chromosomes and if this population of lateral microtubules do not shorten, then the authors may be missing a reduction in the length of the ones that actually are "end-on".

The classification was done by the methods we mentioned earlier in response to reviewer #2 comments. As the chromosomes are round there are also end-on associated microtubules at more peripheral positions on the chromosome surface and not only at the directly pole-facing side of the chromosomes. To support this classification, we have added Video 15 to Materials and Methods. This video shows a pair of autosomes with both populations of microtubules at higher resolution. Our intention was not to show that only end-on associated microtubules contribute to the segregation. We would like to point out that both types of association persist throughout meiosis, and we have changed the text accordingly to reflect this (subsection “Spermatocyte spindles maintain both end-on and lateral associations of

kinetochore microtubules to chromosomes throughout meiosis”).

Related to this, in paragraph four of subsection “Tension release across the spindle may contribute to autosomal anaphase A”, the authors use their conclusion that microtubule lengths are constant at 0.63 µm between metaphase and anaphase to calculate how much altering the microtubule angle could contribute to chromosome movements (ending up with a value of 0.17 µm). However, since this measurement of microtubule length is subject to error (given the variability of microtubule length measurements and the potential issue with distinguishing end-on from lateral interactions), the calculations that led to the 0.17 µm shortening number are also in question.

These measurements (as all measurements) are subject to variability, which we do report. This is simply a theoretical geometric calculation to illustrate how a change in angle, while using the average microtubule lengths, could contribute to a change in pole-to-chromosome length. We do not want to draw absolute conclusions from these theoretical considerations but rather want to show that other mechanisms apart from microtubule shortening could contribute to anaphase A movements.

3) In Figure 10, the authors report the ANOVA comparing metaphase with two of the anaphase spindles (number 3 and 4) and it appears that there is a significant difference, but they do not report the ANOVA comparison between metaphase and anaphase numbers 1 and 2 (and #2 especially does not appear to be different from metaphase). This raises the concern that variability between spindles could account for the different amounts of autosomal stretching, rather than spindle stage (metaphase vs. anaphase). This is particularly important since the authors use this data to propose that release of this stretch can account for much of the "anaphase A" pole-chromosome shortening. To me, it seems problematic to draw these conclusions when only one metaphase spindle is analyzed, and therefore it is difficult to exclude spindle-spindle differences in the autosomal FWHMs.

To address this, we added two additional metaphase data sets. We have also shown all ANOVA analysis in Appendix—figure 5. In the interest of readability of Figure 8, we decided to show all significance tests in the separate Appendix—figure 5.

4) For the analysis of chromosome segregation, it would be helpful to have more information about how the distances were determined. The Experimental Procedures state "For the analysis of chromosome movements, the peak maxima of the chromosome fluorescence signals were then used to calculate the distances for each time point." This is confusing and more details on the analysis should be included so the reader can better evaluate these data – how were the centers of each autosome determined? I can imagine that this would be difficult given the resolution of the videos, and subject to error.

As suggested, we have added more information on how we determined these distances to the Materials and Methods (subsection “Analysis of spindle dynamics”). We indicate that we measured the distances in the resampled data and used a kymograph created along the spindle axis. For that, a radius of 0.9 μm was sampled around the spindle axis in 0.1 μm steps along this axis. The Gaussian weighted sum of fluorescence in each plane was calculated for each time point. The two maxima per time point of this plot were then utilized to determine the distance of the chromosomes.

Also, was each autosome within a given spindle measured separately, or was each segregating mass of autosomes treated as one unit (and the center of that entire mass determined)? Given potential issues in accurately determining the "center" (of either each autosome or the autosome mass), I would suggest that the authors try measuring the distance between the outer edges of the autosomes (the poleward facing sides) and the spindle pole… this would more clearly represent the distance the chromosome travels towards the spindle pole.

As stated above, we measured the distances between the fluorescence intensity peaks of the “segregating masses” for the autosomes. We considered this as the most robust way to determine distances because it utilizes the bulk of chromosomes in the spindle center and therefore eliminates outliers. In the suggested way of measuring the distance from the outer edge, some chromosomes might be not perfectly aligned in the segregating plate. The orientation of the spindle relative to the direction of the z-scan of the 3D imaging setup would also influence accuracy. Further, the distances in the tomographic data were determined by measuring the centers of mass of the individual chromosomes. Therefore, this makes LM and EM measurements also more comparable.

5) The authors state that: "Interestingly, the centrosomes remain connected to the X chromosome-connected microtubules". However, there is no figure call-out for this statement, and I can't find any figures where there is convincing evidence of this, given the resolution of the images/videos presented.

We agree that this paragraph was confusing. We have deleted this whole section.

I am therefore confused as to what evidence supports the subsequent statement "Thus, the separation and migration of centrosomes during anaphase I appears to be coordinated with microtubules that must maintain connections not only to segregating autosomes, but also the lagging X chromosome". Either present these data or revise these statements.

We agree that this statement was confusing. We have taken it out.